# Normal cognitive and social development require posterior cerebellar activity

Aleksandra Badura[1,2,3,4], Jessica L Verpeut[1,3], Julia W Metzger[1,3], Talmo D Pereira[1,3], Thomas J Pisano[1,3,5], Ben Deverett[1,3,5], Dariya E Bakshinskaya[1,3], Samuel S-H Wang[1,3]*

[1]Princeton Neuroscience Institute, Princeton University, Princeton, United States; [2]Netherlands Institute for Neuroscience, Amsterdam, The Netherlands; [3]Department of Molecular Biology, Princeton University, Princeton, United States; [4]Department of Neuroscience, Erasmus MC, Rotterdam, The Netherlands; [5]Robert Wood Johnson Medical School, New Brunswick, United States

**Abstract** Cognitive and social capacities require postnatal experience, yet the pathways by which experience guides development are unknown. Here we show that the normal development of motor and nonmotor capacities requires cerebellar activity. Using chemogenetic perturbation of molecular layer interneurons to attenuate cerebellar output in mice, we found that activity of posterior regions in juvenile life modulates adult expression of eyeblink conditioning (paravermal lobule VI, crus I), reversal learning (lobule VI), persistive behavior and novelty-seeking (lobule VII), and social preference (crus I/II). Perturbation in adult life altered only a subset of phenotypes. Both adult and juvenile disruption left gait metrics largely unaffected. Contributions to phenotypes increased with the amount of lobule inactivated. Using an anterograde transsynaptic tracer, we found that posterior cerebellum made strong connections with prelimbic, orbitofrontal, and anterior cingulate cortex. These findings provide anatomical substrates for the clinical observation that cerebellar injury increases the risk of autism.
DOI: https://doi.org/10.7554/eLife.36401.001

*For correspondence:
sswang@princeton.edu

Competing interests: The authors declare that no competing interests exist.

## Introduction

Human capacities for cognition and flexible behavior unfold rapidly in the first six years of life. During this period, subcortical processing helps refine connections in the developing forebrain (*Knudsen, 2004*; *Wang et al., 2014*; *Wiesel, 1982*). Even though the cerebellum is best known as a structure that guides movement and action (*Dean et al., 2010*), it is also likely to regulate cognitive and emotional processing (*Reeber et al., 2013*; *Snow et al., 2014*), a role that may extend to early development. Cerebellar projections to and from the forebrain are extensive (*Figure 1A*; *Altman and Bayer, 1997*; *Buckner et al., 2011*; *Diamond, 2000*; *Sokolov et al., 2017*; *Wang et al., 2014*) and are present in early life (*Altman and Bayer, 1997*; *Buckner et al., 2011*; *Diamond, 2000*; *Sokolov et al., 2017*; *Wang et al., 2014*). The cerebellum communicates with midbrain and neocortical targets (*Strick et al., 2009*), providing a means for guiding the brainwide maturation of flexible and social behaviors.

Pediatric cerebellar insult causes cognitive and affective deficits (*Limperopoulos et al., 2014*; *Limperopoulos et al., 2010*). Indeed, specific neonatal cerebellar injury increases autism risk by 36-fold (*Limperopoulos et al., 2007*), suggesting that the cerebellum plays a necessary role in cognitive and social development. Finally, in mice, cerebellar-only genetic alterations lead to deficits of flexible and social behavior (*Passot et al., 2012*; *Peter et al., 2016*; *Tsai et al., 2012*).

The cerebellum's role in guiding and shaping behavioral development is likely to be region-specific (*Stoodley et al., 2017*). Anatomical specificity of nonmotor functions is suggested by the

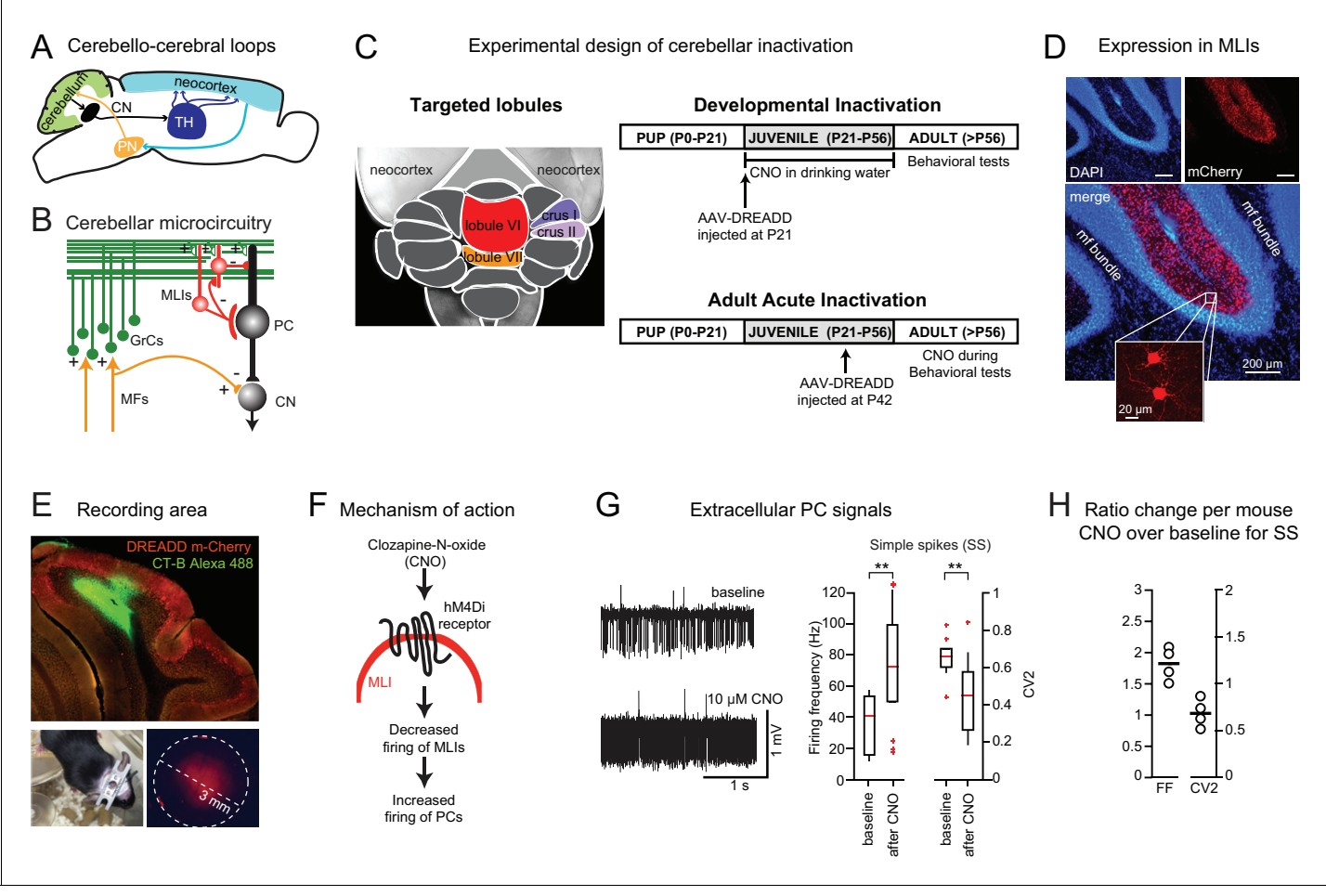

**Figure 1.** Chemogenetic perturbation of cerebellar activity. (**A**) Cerebello-cerebral loops, defined by an ascending pathway via thalamus and a descending reciprocal pathway via pontine nuclei. (**B**) Simplified diagram of cerebellar microcircuitry. Molecular layer interneurons (MLIs) receive incoming excitation from the mossy fiber (MF)-granule cell (GrC) pathway and in turn inhibit Purkinje cells (PCs), the output neurons of the cerebellar cortex which then inhibit neurons of the cerebellar nuclei (CN). (**C**) *Left,* dorsal view of cerebellum with the four targeted lobules indicated in color. *Right,* experimental design for developmental and acute perturbation. (**D**) Expression of the chemogenetic DREADD probe hM4D(Gi)-mCherry in MLIs (red). Note the absence of mCherry signal in the granule cell layer or the mossy fiber bundle visualized by DAPI staining (blue) (see *Video 1*). (**E**) *Top,* a sagittal cerebellar section showing an example recording location in the in vivo awake experiment. The recording location was marked by cholera toxin subunit B conjugated to Alexa 488 staining (green); DREADD expression marked by mCherry (red). *Bottom left,* removable implant used for in vivo electrophysiology. *Bottom right,* mCherry expression imaged through the implant silicone plug. (**F**) The activating ligand clozapine-N-oxide (CNO) binds to the hM4Di receptor, which decreases firing of MLIs (see *Figure 1—figure supplement 1*) and thus removes synaptic inhibition from PCs. (**G**) *Left,* extracellular recording of PC activity from awake mice before and after CNO application. *Right,* CNO (10 µM) leads to an increase in the simple-spike firing frequency and a decrease in the local coefficient of variation (CV2). **, different from baseline by paired t-test, $p < 0.05$ (**H**) CNO-to-baseline ratios of the measures, plotted on a cell-by-cell basis.

DOI: https://doi.org/10.7554/eLife.36401.002

The following figure supplement is available for figure 1:

**Figure supplement 1.** CNO administration alters cerebellar activity in vitro.

DOI: https://doi.org/10.7554/eLife.36401.003

existence of cerebellar microzones, which contain repeating stereotypical circuit motifs (*Figure 1B*) and generate a systematic mediolateral map that projects in a characteristic fashion to the deep nuclei, the output structures of the cerebellum. In addition, the cerebellar cortex is heterogeneous along the anteroposterior axis, projecting to midbrain and neocortical targets via organized anatomical pathways, and receiving substantial descending inputs from the same structures to which they project (*Strick et al., 2009*). This anteroposterior organization is typically categorized by lobules, which provide defined targets for anatomical mapping and functional perturbation. Overall,

cerebellar connections form a bidirectional map, not only to sensorimotor regions, but also to cognitive and affective areas (*Koziol et al., 2014*; *Popa et al., 2014*; *Wang et al., 2014*).

Together, these previous findings suggest that the cerebellum plays a crucial role in the developmental maturation and adult expression of flexible and social behaviors. We tested this hypothesis using three tools. First, we used Designer Receptors Exclusively Activated by Designer Drugs (DREADDs) to achieve reversible, anatomically-localized perturbation of a specific cell class in freely moving animals. DREADDs do not act unless exposed to an activator molecule (*Wess et al., 2013*). Using this approach, we could reversibly perturb neural function in adult or juvenile mice in individual lobules for minutes to days, test adult behavioral outcomes, and recover the spatial distribution of DREADD expression. Second, we monitored behavioral alterations using a variety of assays to identify patterns that span multiple measures and even multiple tasks. Such experiments require detailed quantitative analysis because individual tasks combine motor and nonmotor capacities (*Crawley, 2007*). Third, we used transsynaptic tracing viruses to identify forebrain regions likely to contribute to the observed effects. These experiments allowed us to interrogate lobule-specificity, behavioral consequences, and distal anatomical targets of cerebellar influence.

## Results

### Experimental design

To probe the role of identified cerebellar regions (*Figure 1C*) during cognitive and social development, we manipulated neural activity in mice reversibly using Designer Receptors Exclusively Activated by Designer Drugs (DREADDs; *Wess et al., 2013*). We injected adeno-associated virus (AAV) carrying the sequence for the inhibitory DREADD hM4Di, fused to mCherry protein under a synapsin-1 promoter (*Kuhn et al., 2012*) to drive expression exclusively in molecular layer interneurons (MLIs; *Figure 1D* and *Video 1*; of lobules VI or VII, crus I or II, or paravermal lobule VI). Mice underwent a battery of behavioral testing and training, followed by recovery of the distribution of DREADD expression by two-photon fluorescence tomography.

As our developmental perturbation, after AAV injection at postnatal day (PND) 21 (*Figure 1C*; *Table 1*), we administered the DREADD agonist clozapine-N-oxide (CNO) on PND 30 – 56. To compare developmental effects with the direct effects of adult disruption of activity, we performed additional experiments in which we injected virus at PND 42 – 48 and tested the acute effects of CNO administered prior to behavioral testing (*Figure 1C*; *Table 1*). All behavioral tests were done between PND 57 and PND 126.

Activation of inhibitory DREADDs in MLIs should affect cerebellar Purkinje cell output in two major ways, by disinhibiting simple spiking and by impairing modulation of spike rate and timing (*Figure 1E–H*). These alterations would, in turn, suppress the amount and modulation of deep-nuclear output to the rest of the brain. Using in vivo extracellular single-unit recording, we confirmed that CNO administration induced cell-specific increases in Purkinje cell simple-spike firing frequency (FF) and a reduction in the coefficient of variation for a sequence of two interspike intervals (CV2) (n = 4 mice, p=0.047 for FF and p=0.024 for CV2, paired t-test, effect size: Cohen's *d* = 1.5 pooled standard deviations for FF and 1.3 for CV2) (*Figure 1G–H*). Furthermore, we confirmed effects on MLIs in cerebellar brain slices (*Figure 1—figure supplement 1*).

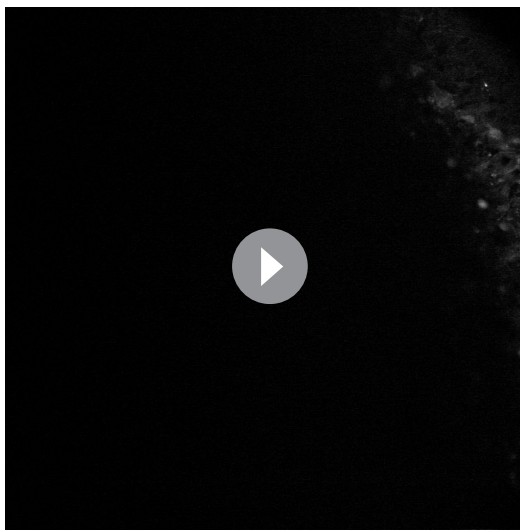

**Video 1.** Dense expression of hM4D(Gi)-mCherry in molecular layer interneurons.
DOI: https://doi.org/10.7554/eLife.36401.004

**Table 1.** Summary of all mice.

| Experimental group | Injection site | Behavioral assays | | | | | |
|---|---|---|---|---|---|---|---|
| | | Grooming | Social chamber | Y-maze | Elevated Plus Maze | Gait | Eyeblink |
| Adult | Lobule VI | 12 | 12 | 12 | 12 | 12 | 8 |
| | Lobule VII | 10 | 10 | 7* | 10 | 10 | - |
| | Crus I | 10 | 10 | 10 | 10 | 10 | 8 |
| | Crus II | 11 | 11 | 11 | 11 | 11 | 6 |
| | Eyeblink | - | - | - | - | - | 9 |

*not counting three mice excluded from group comparison during habituation phase

| | | | | | | | |
|---|---|---|---|---|---|---|---|
| Developmental | Lobule VI | 13 | 13 | 13 | 13 | 13 | 6** |
| | Lobule VII | 8 | 8 | 8 | 8 | 8 | - |
| | Crus I | 7 | 7 | 7 | 7 | 7 | 6** |
| | Crus II | 12 | 12 | 12 | 12 | 12 | 3 |

**not counting four mice (1 crus I, 3 lobule VI) excluded due to eyeblink-zone spillover expression

| Controls | Type | | | | | | |
|---|---|---|---|---|---|---|---|
| Adult | DREADDs (eyeblink zone)+saline injections | - | - | - | - | - | 5 |
| | Saline i.p. | 9 | 9 | 9 | 9 | 9 | - |
| | CNO alone | 10 | 10 | 10 | 10 | 10 | 9 |
| | No treatment | 8 | 8 | 17 | 18 | 18 | 16 |
| | GCaMP6f injections + CNO | - | - | 8 | - | - | - |
| Developmental | CNO alone | 10 | 10 | 10 | 10 | 10 | - |

DOI: https://doi.org/10.7554/eLife.36401.005

## Region-specific impairment of eyeblink conditioning, but minimal effects on gait

We tested whether DREADD activation could affect classical eyeblink conditioning, a cerebellum-dependent associative learning task (*Figure 2*). By pairings of light flashes (conditioned stimulus, CS) with airpuffs to the cornea (unconditioned stimulus, US) mice were trained to preemptively close the eye in response to the light alone, a learned behavior termed a conditioned response (CR; *Figure 2A*). DREADD activation during development in either paravermal lobule VI ('eyeblink area') or crus I, both of which modulate eyeblink conditioning (*Giovannucci et al., 2017*; *Heiney et al., 2014*), was sufficient to cause long-lasting deficits in learning (*Figure 2B*), suggesting that normal activity in these regions is necessary for the maturation of eyeblink conditioning (*Freeman, 2014*). The necessity of these regions for conditioning was confirmed by acute inactivation (*Figure 2C*). This effect was reversible (*Figure 2D*). No impairments were found with CNO or DREADD treatment alone (*Figure 2—figure supplement 1A*) or from perturbation of other lobules (*Figure 2—figure supplement 1B–C*). Thus, DREADD-based perturbation can cause lobule-specific alterations in the development and adult expression of cerebellum-dependent associative learning.

To test for cerebellar influences on motor capacity that might affect tasks, we analyzed gait (*Machado et al., 2015*); *Figure 3A–B*; *Videos 2* and *3*) and found that individual posterior cerebellar lobules were not necessary for the expression of normal gait parameters. As a yardstick of the full scale of cerebellum-specific impairment of gait, we used Purkinje cell-specific $Tsc1^{-/-}$ mice, which show cerebellar degeneration and ataxia (*Tsai et al., 2012*); $L7^{Cre};Tsc1^{flox/flox}$). $Tsc1^{-/-}$ mice showed considerably wider forelimb and hindlimb stance than controls (forelimb/hindlimb effect size $d$ = 2.4, *Figure 3C*, red dots). This result was consistent with previous measurements in *Agtpbp1* mice (see *Figure 3B* in *Machado et al., 2015*). Compared with $Tsc1^{-/-}$ mice, CNO treatment by itself in wild-

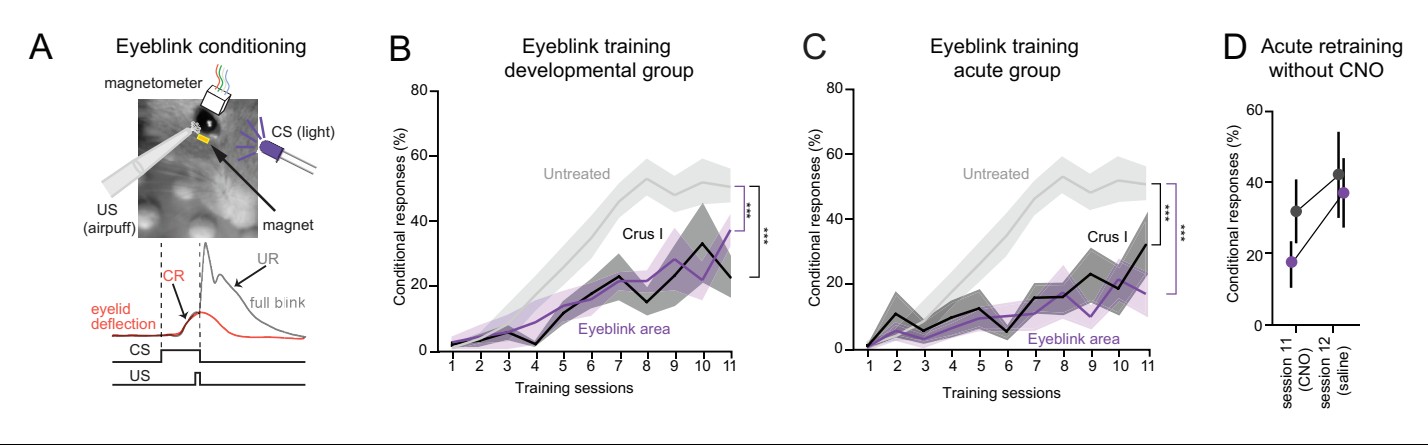

**Figure 2.** Juvenile-life perturbation disrupts the development of cerebellar-dependent eyeblink conditioning. (A) *Top*, eyeblink conditioning performed using an unconditional stimulus (US, corneal airpuff) delivered at the end of a conditional stimulus (CS, LED). *Bottom*, learned anticipatory eyelid deflection (conditional response, CR; red), followed by an unconditional reflex (UR) blink. (B) Reduced frequency of CRs after developmental CNO activation of DREADDs in the eyeblink area of lobule VI ($p<10^{-10}$, two-way ANOVA) and in crus I ($p<10^{-6}$) compared to controls (see ***Figure 2—figure supplement 1***). (C) Reduced frequency of conditional responses after acute CNO activation of DREADDs in lobule VI eyeblink area ($p<10^{-10}$, two-way ANOVA) and in crus I ($p<10^{-6}$). (D) In adult-disrupted mice, removal of CNO after 11 sessions of training resulted in recovery of conditional responses in both affected acute groups (crus I and eyeblink region). Error bars show mean ±SEM.

DOI: https://doi.org/10.7554/eLife.36401.006

The following figure supplement is available for figure 2:

**Figure supplement 1.** Blockade of eyeblink conditioning by DREADD activation in eyeblink-relevant regions.

DOI: https://doi.org/10.7554/eLife.36401.007

type mice had smaller effects (acute *d* = 0.2, developmental *d* = 1.0), potentially arising from its conversion to a bioactive product (***Gomez et al., 2017***; ***Manvich et al., 2018***). Using CNO-only controls as a comparison, we found that in virus-injected mice, CNO treatment led to small changes in stance (acute *d* = −0.3 to +0.7, developmental *d* =−0.2 to +0.3; ***Figure 3D***, ***Video 3***). With the exception of lobule VII (***Figure 3D***), acute and developmental perturbation of posterior cerebellar lobules did not cause statistically significant gait deficits.

## Developmental CNO exposure leads to subtle behavioral deficits

To probe cognitive and affective function, we administered four behavioral tests (Y-maze, social chamber, grooming assay and elevated plus-maze) to obtain a panel of quantitative information (***Table 2***). Acute CNO administration in virus-untreated mice had no detectable effects on any behavioral assay (***Figures 4–8***). Developmentally, in virus-untreated mice, we found that CNO administration alone did not have detectable effects on Y-maze learning or social chamber behavior (***Figures 4–6***; p>0.05 for all groups for all metrics), but it did have modest effects on elevated plus-maze (EPM) parameters (***Figure 7***) and self-grooming (***Figure 8***). For further analysis of treatment effects, baseline control groups were age-matched and virus-untreated: for Y-maze and social chamber, no-CNO and CNO-alone mice combined; and for EPM and grooming, CNO-alone mice.

## Nonmotor deficits in multiple behavioral tests following developmental cerebellar inactivation

### Y-maze reversal learning

To test flexible learning, we used a swimming Y-maze in which mice were habituated to a maze, initially taught to find an underwater platform in one maze arm, and later switched to the other arm (***Figure 4A***). Initial learning took place over four sessions (5 trials each, day 1, 'Acquisition'), followed by a test for retention (day 2, 'Test'). All control groups, including CNO acute and developmental exposure, virus-uninjected mice, and GFP expression controls, performed above chance overall on the first day of initial training (***Figure 4B***; ***Figure 4—figure supplement 1***) and reached near-perfect performance by the third session. Three mice from the adult lobule VII injection group were

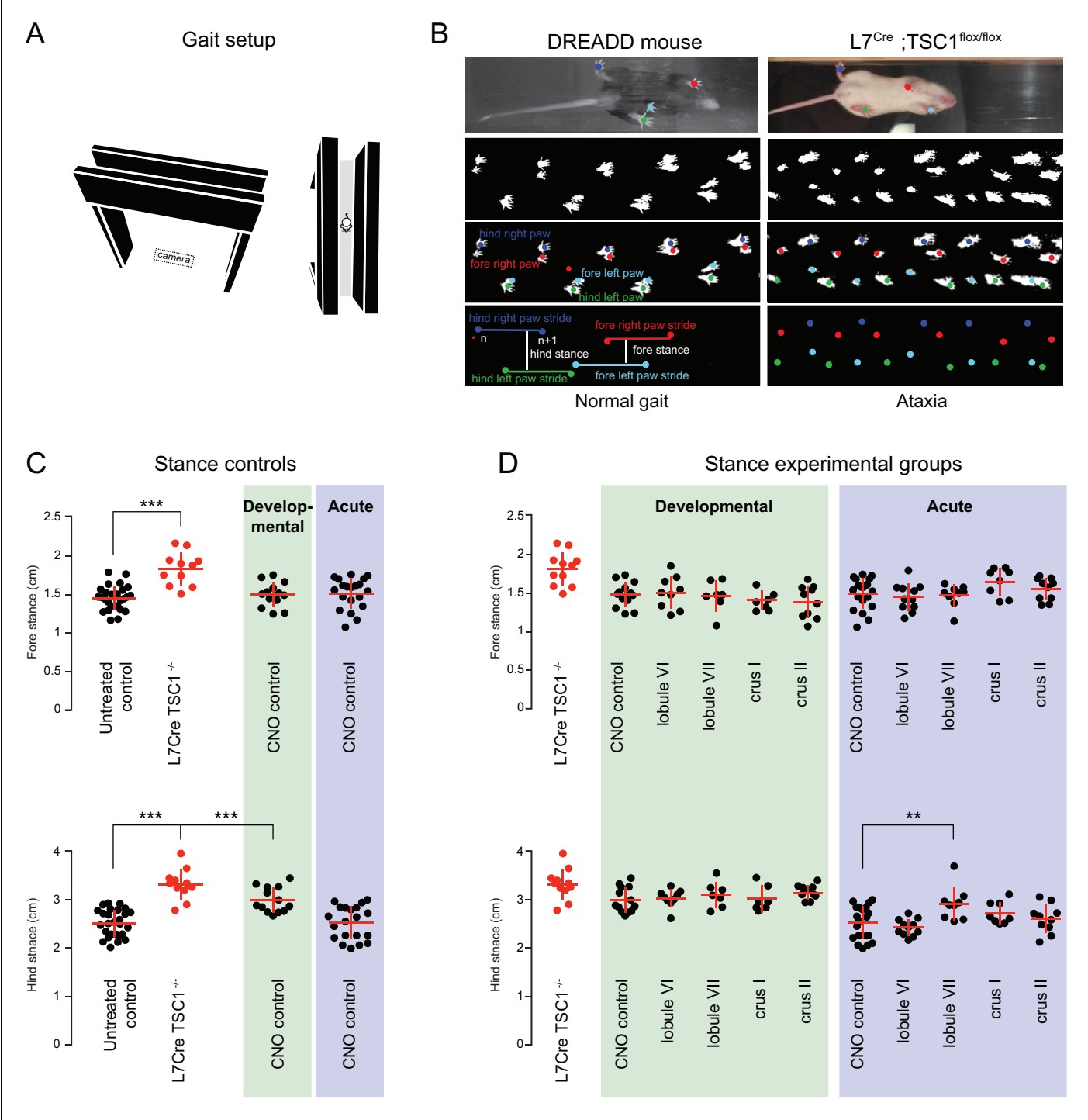

**Figure 3.** Mild gait impairments arise from prolonged CNO exposure. (**A**) Schematic of gait apparatus. A camera was placed below the plexiglas to record a mouse running the length of the track. (**B**) Raw gait videos were processed using a MATLAB graphical user interface to detect sequences of mouse locomotion. *Left*, images from a mouse expressing DREADDs in lobule VII after acute CNO administration. *Right*, images from an ataxic L7[Cre]; Tsc1[flox/flox] mutant adult mouse used as a positive control. Stride lengths for each paw were calculated as the average distance between successive paw placements (see *Video 2*). Fore and hind stances were calculated as the average distance between fore and hind paws measured in the direction of the locomotion. (**C**) Stance measurements revealed no acute (blue shading) or developmental (green shading) effects of CNO on fore stance when compared to untreated controls (no shading). L7[Cre];Tsc1[flox/flox] mice were used as positive controls (red). Developmental exposure to CNO resulted in an broader hind stance (CNO control developmental vs. control p<0.0001, one-way ANOVA). Hind stance in the L7[Cre];Tsc1[flox/flox] mice was more
*Figure 3 continued on next page*

*Figure 3 continued*

severely affected (p<10$^{-9}$). (**D**) Acute perturbation to lobule VII caused a mild broadening of hind (but not fore) stance when compared to CNO exposure time matched control (p<0.01, one-way ANOVA, see **Video 3**).

DOI: https://doi.org/10.7554/eLife.36401.008

excluded during the habituation phase because they had difficulty swimming, possibly arising from the cutting of neck muscles in adult life required in surgery (see Materials and methods).

For all lobules, DREADD-injected mice treated in juvenile life with CNO learned at the same rate as uninjected mice (**Figure 4C**). Acute exposure to CNO in adult DREADD-injected mice resulted in no impairment in the rate of acquisition except for a small impairment in lobule VI-injected mice (**Figure 4D**, left panel).

To test reversal learning, we next moved the platform to the opposite arm and trained mice to learn this new location for 2 days (days 3 and 4, 'Reversal 1' and 'Reversal 2', **Figure 4A**). All developmental-perturbation groups showed significant reversal on day three except for lobule VI mice, which lagged behind other groups (**Figure 4C**; repeated-measures ANOVA, p<0.001, Dunnett's multiple comparisons post-hoc test p<0.01; *d* = 1.1 for multisession reversal 1). Crus I developmental mice also showed decreased flexible behavior as measured by multisession reversal one and final reversal 1 (both *d* = 0.6; **Figure 4C** and **Table 3** and **Table 3—source data 1**), although this measure did not reach statistical significance.

At this point, for the fifth session of day 3, the incorrect arm was blocked, thus leaving mice only the correct arm as an available path (see **Figure 4A**). Nevertheless, lobule VI mice persisted on the second day of reversal training (day 4, 'Reversal 2'; repeated-measures ANOVA, p<0.05, Dunnett's multiple comparisons post-hoc test p<0.05; *d* = 0.4, initial reversal 2, **Table 3—source data 1**). Similar, smaller impairments in reversal learning were seen with acute disruption of lobule VI (**Video 4**; **Figure 4D**). In summary, developmental and to a lesser extent acute disruption of activity in cerebellar lobule VI led to perseveration and, in the case of development, a lasting failure of reversal learning.

Notably, the impairment in reversal learning was not coupled with decreased mobility as measured by distance swum during habituation. Both developmental and acute groups performed similarly to the control groups (**Figure 5A**), except that acute perturbation to lobule VI resulted in a slight increase in swimming distance (repeated-measures ANOVA, p<0.05, Šidák's multiple comparisons post-hoc test p<0.05).

To further analyze behavioral variation in the Y-maze, we analyzed individual metrics of performance and learning (**Figure 5**). Developmental perturbation had no detectable effect on pre-training swimming distance (**Figure 5A**). However, detailed specific measures of both day 1 initial-phase learning and day 3 reversal learning (**Figure 5B**) were substantially impaired (**Table 3** and **Table 3—source data 1**).

If conditions that impaired reversal learning had effects that were specific to learning, those conditions might be expected to induce variability in specific learning metrics. To test this idea, we calculated the variance in various Y-maze parameters and normalized them to the control group for comparison (**Figure 5C**). We found that the experimental groups did not have increased variance in distance swum (F-test; p=0.15), but did have increased variance in multisession reversal 1 (p=0.005) and final reversal 1 (p=0.006).

To test whether the changes in Y-maze performance followed a pattern that spanned multiple behavioral measures, we used principal component analysis (PCA) (**Figure 5—figure supplement 1A**). PCA identifies variation that occurs in a concerted manner among multiple behavioral parameters, a pattern that pairwise

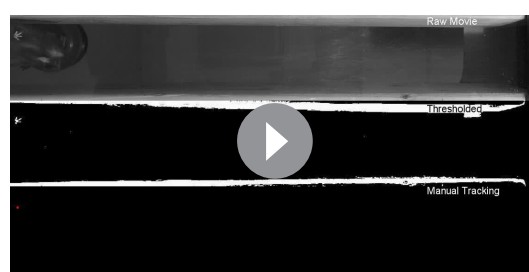

**Video 2.** Example of gait analysis. *Top.* Raw movie played at four fps. *Middle.* Thresholded movie. *Bottom.* Tracked paws using 'Manual Tracker' plugin in Fiji.
DOI: https://doi.org/10.7554/eLife.36401.009

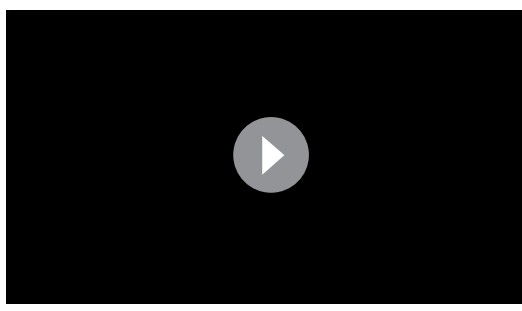

**Video 3.** Example of gait measurement.
DOI: https://doi.org/10.7554/eLife.36401.010

correlations between individual parameters do not capture (*Jolliffe, 2014*). We used pooled results for untreated and CNO-only mice (n = 36) to construct a control behavioral space. (This same group of mice was also used to calculate PCA for the social chamber task.) We projected each perturbation group's behavioral measures onto the control group's principal component (PC) space.

The first PC ('PC1'), which by definition captured the most variance in the control group, was dominated by distance measures (*Figure 5—figure supplement 1B*). In contrast, in experimental groups the highest-contribution PCs (see Materials and methods) were consistently different from PC1 (*Figure 5—figure supplement 1C–D*). PCs recaptured the differences in variance observed among the behavioral metrics (*Figure 5C*). The experimental groups were distinct from controls in the amount of variation captured by reversal-learning-containing components PC3 and PC5, reflecting reversal learning (*Figure 5D*; in a unit vector, multisession reversal one has a coefficient weight of 0.68 in PC3; final reversal one has coefficient weight 0.48 in PC5). In developmental lobule VI mice, 41% of total variance arose from PC3 and PC5. In developmental crus I, PC5 captured 18% of the variance. Taken together, PCA analysis indicates that the dominant consequence of developmental perturbation in Y-maze performance was non-locomotor in nature and largely independent of movement parameters.

## Social behavior

To probe social interaction, we monitored mice in a three-chambered apparatus (*Figure 6*) in which a novel mouse and object were simultaneously presented in opposite side chambers (*Figure 6A*). The test mouse was placed in the middle chamber and its actions recorded by camera and analyzed using an automated tracking system. Interactions with the novel mouse or object were tabulated if the test mouse was in proximity to the cup (*Figure 6B* and *Table 2*; see Materials and methods for details).

All mice, uninjected and lobule-injected, spent more time in side chambers if they contained a novel mouse or object than if the side chambers were empty (significant sides-over-center preference for mouse/object greater than sides-over-center preference with all chambers empty, $p=5 \times 10^{-6}$ to 0.006, paired t-test). Left/right chamber preference was uncorrelated with preference under baseline conditions (i.e. trials without object or mouse), suggesting an absence of chamber bias arising from environmental factors such as odor or landmarks (*Figure 6—figure supplement 1A*).

Disruption of crus I or crus II during juvenile life led to profound adult indifference between mouse and object as measured by time spent in the mouse chamber and in close proximity to the cup (*Figure 6C*, mouse vs object p=0.85 for crus I and p=0.5 for crus II). All control groups showed a mouse-over-object preference during the test phase (*social preference*, parameter #9; *Figure 6C*, two-way ANOVA, p<0.0001, Šidák's multiple comparisons post-hoc test; $d$ = 3.5). There were no differences among control groups for any of the social chamber parameters (*Figure 6—figure supplement 1B*).

Changes in preference for mouse over object were not seen in adult-injected mice for any lobule, 20 min after administration of CNO (*Figure 6C*, two-way ANOVA, p<0.0001, Šidák's multiple comparisons post-hoc test p<0.01 for all lobules; $d$ = 2.8 to 4.9), with no statistically detectable differences between them or with uninjected groups (one-way ANOVA, p=0.2). Thus, the capacity of mice to express social preferences in the three-chamber apparatus did not depend acutely on activity in any cerebellar region tested, lobule VI or VII or crus I or II.

In contrast, developmental perturbation did disrupt social preference. Developmentally induced indifference to social stimuli was largely not accompanied by decreases in movement (*Figure 6—figure supplement 1C*; p>0.05, two-tailed for distance traveled during either baseline or test period,

**Table 2.** Definitions of behavioral metrics.

| # | Measure | Description | Quantification | Direction indicating impairment |
|---|---------|-------------|----------------|--------------------------------|
| | | Elevated Plus Maze metrics (EPM_) | | |
| 1 | Commitment | Full entrances to the open arms ($\mathrm{Entr_{Ofull}}$) relative to sum of full entrances and jittery entrances to the open arms ( $\mathrm{Entr_{Ojitter}}$) | $\dfrac{\mathrm{Entr_{Ofull}}}{\mathrm{Entr_{Ofull}}+\mathrm{Entr_{Ojitter}}}$ | experimental < controls |
| 2 | Distance | Distance traveled (cm) during the ten minutes in the Elevated Plus Maze | $\mathrm{EPM_{distance}}$ | experimental < controls |
| 3 | Exploration Entrances | Entrances into the crossroads central area | $\mathrm{Entr_{central}}$ | experimental < controls |
| 4 | Exploration Time | Time in the crossroads central area | $\mathrm{Time_{central}}$ | experimental < controls |
| 5 | Open-Arm Preference | Time in the open arms ($\mathrm{Time_{open}}$) relative to total time in closed ($\mathrm{Time_{closed}}$)and open arms | $\dfrac{\mathrm{Time_{open}}}{\mathrm{Time_{closed}}+\mathrm{Time_{open}}}$ | experimental < controls |
| | | Grooming metrics (GR_) | | |
| 6 | Grooming Ratio | Difference betweenaverage grooming bout length in CNO ($\mathrm{AVG_{CNO}}$) and SALINE condition ($\mathrm{AVG_{saline}}$) relative to SALINE condition | $\dfrac{\mathrm{AVG_{saline}}-\mathrm{AVG_{CNO}}}{\mathrm{AVG_{saline}}}$ | experimental > controls |
| | | Social Chamber metrics (SC_) | | |
| 7 | Baseline Distance | Distance traveled (m) during baseline phase (10 min free exploration of the empty social chamber apparatus) | $\mathrm{BS_{distance}}$ | experimental < controls |
| 8 | Novelty-Seeking | Difference between summed entrances to mouse ($\mathrm{Entr_{M}}$) and object ($\mathrm{Entr_{O}}$) chambers in test ($_{test}$) and baseline ($_{bs}$) sessions, relative to baseline session | $\dfrac{((\mathrm{Entr_{Mbs}}+\mathrm{Entr_{Obs}})-(\mathrm{Entr_{Mtest}}+\mathrm{Entr_{Otest}}))}{(\mathrm{Entr_{Mbs}}+\mathrm{Entr_{Obs}})}$ | experimental < controls |
| 9 | Social Preference | Time spent interacting with the novel mouse ($\mathrm{Time_{NearM}}$) relative to total time interacting with either the novel mouse or novel object ($\mathrm{Time_{NearO}}$) | $\dfrac{\mathrm{Time_{NearM}}}{\mathrm{Time_{NearM}}+\mathrm{Time_{NearO}}}$ | experimental < controls |
| 10 | Test Distance | Distance traveled (m) during test phase (10 min exploration of the social chamber apparatus with the novel mouse and object present) | $\mathrm{Test_{distance}}$ | experimental < controls |
| | | Y-maze metrics (YM_) | | |
| 11 | Final Learning | Mean of the percent correct trials in acquisition sessions 3 ($\mathrm{ACQ_{S3}}$) and 4 ($\mathrm{ACQ_{S4}}$) | $\dfrac{\mathrm{ACQ_{S3}}+\mathrm{ACQ_{S4}}}{2}$ | experimental < controls |
| 12 | Initial Learning | Percent correct trials in acquisition session 1 ($\mathrm{ACQ_{S1}}$) | ($\mathrm{ACQ_{S1}}$) | experimental < controls |
| 13 | Multisession Learning | Slope of the linear regression of acquisition sessions 1 ($\mathrm{ACQ_{S1}}$), 2 ($\mathrm{ACQ_{S2}}$), and 3 ($\mathrm{ACQ_{S3}}$) | linear regression slope of ($\mathrm{ACQ_{S1}}$; $\mathrm{ACQ_{S2}}$; $\mathrm{ACQ_{S3}}$) | experimental < controls |
| 14 | Distance | Combined distance swum (m) in the three habituation trials (HAB1, HAB2 and HAB3) (60 s each) of free swimming in the empty Y-maze apparatus | $\mathrm{HAB1_{distance}}+\mathrm{HAB2_{distance}}+\mathrm{HAB3_{distance}}$ | experimental < controls |
| 15 | Final Reversal 1 | Mean of the percent correct trials in reversal day 1 sessions 3 ($\mathrm{RD1_{S3}}$) and 4 $\mathrm{RD1_{S4}}$ | $\dfrac{\mathrm{RD1_{S3}}+\mathrm{RD1_{S4}}}{2}$ | experimental < controls |
| 16 | Initial Reversal 1 | Percent correct trials in reversal day 1 session 1 ($\mathrm{RD1_{S1}}$) | $\mathrm{RD1_{S1}}$ | experimental < controls |
| 17 | Multisession Reversal 1 | Slope of the linear regression of reversal day 1 sessions 1 ($\mathrm{RD1_{S1}}$), 2 ($\mathrm{RD1_{S2}}$) and 3 ($\mathrm{RD1_{S3}}$) | linear regression slope of ($\mathrm{RD1_{S1}}$; $\mathrm{RD1_{S2}}$; $\mathrm{RD1_{S3}}$) | experimental < controls |
| 18 | Final Reversal 2 | Mean of the percent correct trials in reversal day 2 sessions1 ($\mathrm{RD2_{S1}}$), 2 ($\mathrm{RD2_{S2}}$), 3 ($\mathrm{RD2_{S3}}$) and 4 ($\mathrm{RD2_{S4}}$) | $\dfrac{\mathrm{RD2_{S1}}+\mathrm{RD2_{S2}}+\mathrm{RD2_{S3}}+\mathrm{RD2_{S4}}}{4}$ | experimental < controls |
| 19 | Initial Reversal 2 | Percent correct trials in reversal day 2 session 1 | $\mathrm{RD2_{S1}}$ | experimental < controls |

DOI: https://doi.org/10.7554/eLife.36401.022

except for an increase in distance during baseline for crus I). To test whether alterations in social behavior arose from the amount of movement, individual performance parameters were analyzed (*Figure 6D*). Compared with control mice, experimental groups did not show an increase in the variance of baseline or test distance (F-test; p=0.43 for baseline and p=0.17 for test distance). However,

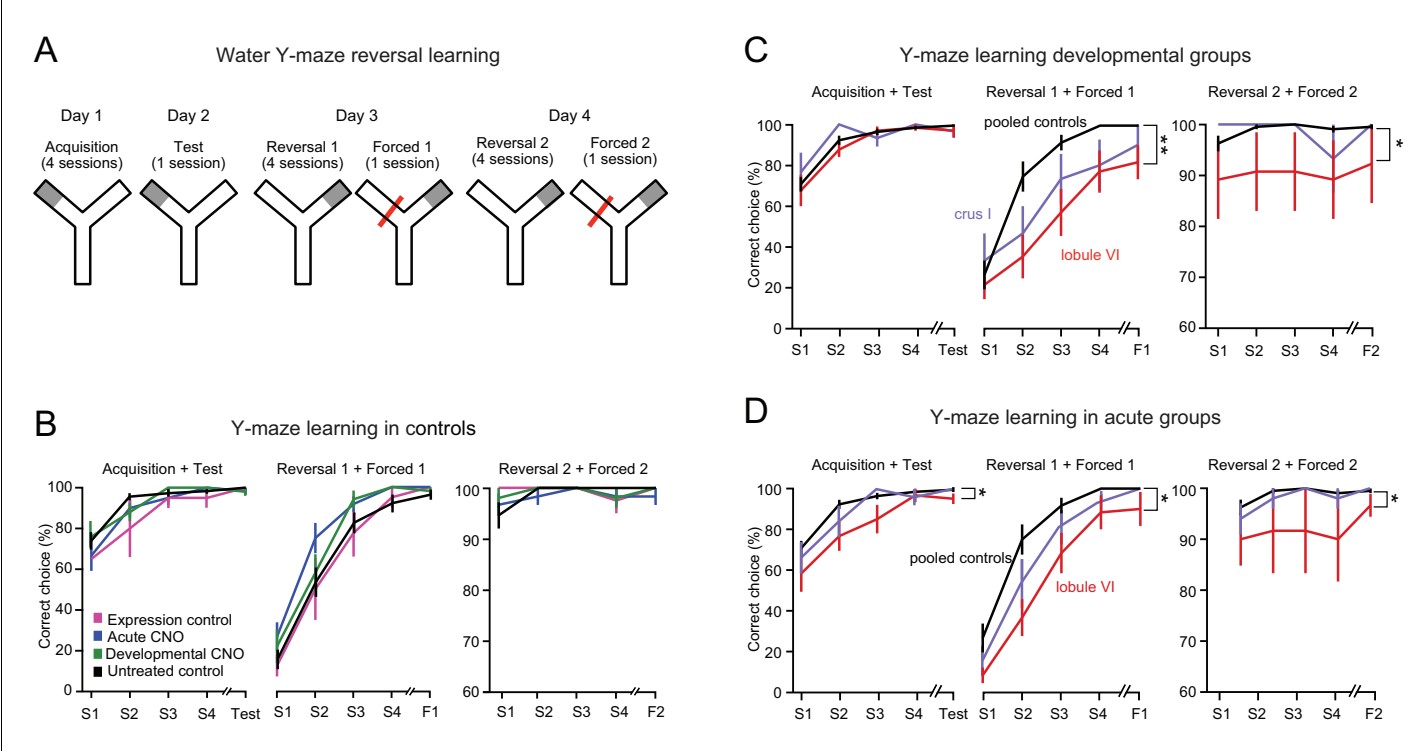

**Figure 4.** Impairment of swimming Y-maze reversal learning. (A) Protocol for the Y-maze reversal assay consisting of initial training (day 1), test (day 2), and reversal (days 3 and 4). (B) All control groups showed normal reversal learning. Data represent averages of all control mice segregated by the type of controls. (C) Reversal learning was impaired by developmental activation of DREADDs in lobule VI and crus I but not in control mice (see *Figure 4— figure supplement 1* and *Videos 4* and *5*). (D) Reversal learning was impaired by acute activation of DREADDs in lobule VI. Data with error bars are plotted as mean ±SEM. *p<0.05; **p<0.01.
DOI: https://doi.org/10.7554/eLife.36401.011

The following figure supplement is available for figure 4:

**Figure supplement 1.** Control distributions of y-maze metrics.
DOI: https://doi.org/10.7554/eLife.36401.012

novelty-seeking and social preference showed increased variance (F-test; p=0.02 for novelty-seeking and p<0.001 for social preference).

To probe covariations in social-chamber behavioral parameters, PCA analysis was done in the same manner as the Y-maze analysis. Just as in the Y-maze, the first PC ('PC1') for social chamber control groups was dominated by distance measures (*Figure 6—figure supplement 2A*). In experimental groups, PC3 was weighted highly on social preference and novelty-seeking, capturing 60% of the behavioral variance after developmental disruption of crus I, and 48% of variance after developmental disruption of crus II (*Figure 6—figure supplement 2B and C*). In addition, PC2 and PC3 together captured 36% of the variance arising from developmental disruption of lobule VII. The effects captured by PC3 were in opposite directions for developmental and adult lobule VII disruption, thus separating the two groups better than any individual measure (*Figure 6E*). In summary, lobule VII activity co-regulates social preference and novelty-seeking in a cohesive manner, and the direction of that influence goes in one direction in juvenile development but reverses in mature adults.

## Elevated plus-maze (EPM)
To probe novelty preference in a non-social context, we placed mice in an EPM with open and closed arms (*Figure 7A–B*), which probes a variety of capacities, including anxiety and exploratory behaviors (*Holmes et al., 2000*). Because the elevated-plus maze has a strong locomotor component (*Wall and Messier, 2000*), measures were designed that normalized open and closed arm

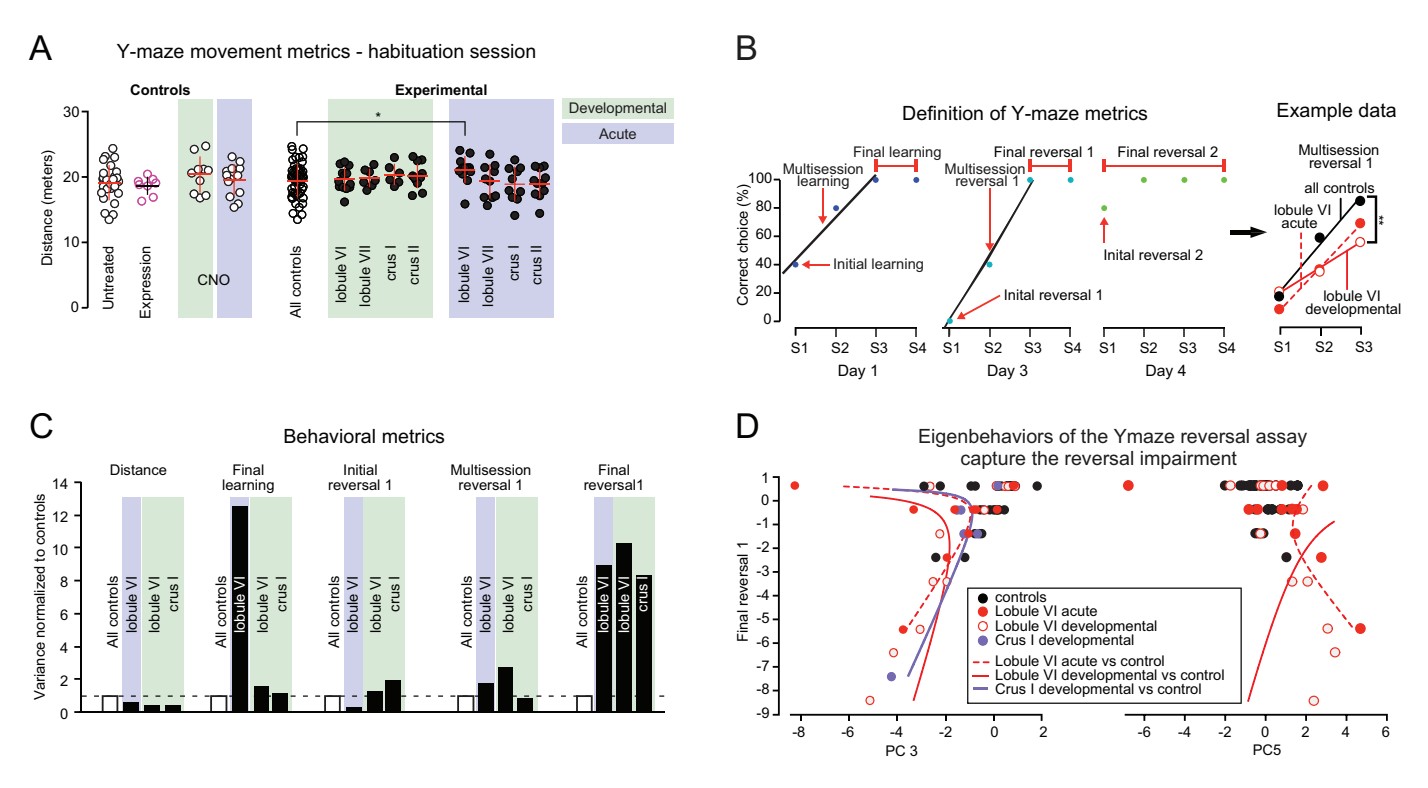

**Figure 5.** Specific impairment of learning parameters in the swimming Y-maze. (**A**) Distance swum by control and experimental groups during the habituation phase of the swimming Y-maze. The only change observed was a mild increase after acute perturbation of lobule VI (p<0.05, one-way ANOVA). (**B**) Nonmotor, learning-related performance measures (for exact definitions see *Table 2*). (**C**) In experimental groups, variance in behavioral metrics is greater for learning metrics than for movement. (**D**) Principal component analysis (see *Figure 5—figure supplement 1*) reveals eigenbehaviors that capture concerted nonmotor impairments in lobule VI acute, lobule VI development, and crus I development groups.
DOI: https://doi.org/10.7554/eLife.36401.013

The following figure supplement is available for figure 5:

**Figure supplement 1.** Dimensionality reduction of behavioral metrics and PCs of the Y-maze assay.
DOI: https://doi.org/10.7554/eLife.36401.014

activity to overall movement (see Materials and methods). Juvenile-injected lobule VII mice showed enhanced closed-arm preference (*Figure 7C*, *d* = 0.9 Šidák's multiple comparisons post-hoc test p=0.02), as well as decreased exploration time (*Figure 7D*, one-way ANOVA, p<0.01, *d* = 1.7, Šidák's multiple comparisons post-hoc test p=0.02) and distance (*Figure 7—figure supplement 1B*, p<0.01, *d* = 1.2; Šidák's multiple comparisons post-hoc test p=0.04). The effect on closed-arm preference was opposite to that seen in adult disruption, which enhanced open-arm preference (*Figure 7C*) and decreased total movement in the EPM for crus II (*Figure 7—figure supplement 1B*).

To further probe the developmental contribution of lobule VII to novelty-seeking, we returned to the three-chamber social test data. Despite the fact that developmental CNO exposure did not detectably alter mouse-over-object preference, it did reduce total novelty-seeking (*Figure 7—figure supplement 2*, *d* = 1.1 two-tailed t-test p<0.01; *Table 2*). This effect was opposite to that seen in adult disruption, which led to increases in novelty-seeking (*Figure 7—figure supplement 2*). Thus, lobule VII co-regulates both non-social and social novelty-seeking in a concerted, development-specific manner.

Juvenile perturbation of lobule VII was also the only condition to prolong grooming events (*Figure 8*). Mice were injected with saline on day 1 and CNO on day 2 (*Figure 8A*). Average grooming events tended to be shorter in mice that had received CNO treatment in drinking water during development even with no DREADDs (*Figure 8B*). Therefore all comparisons were made to the

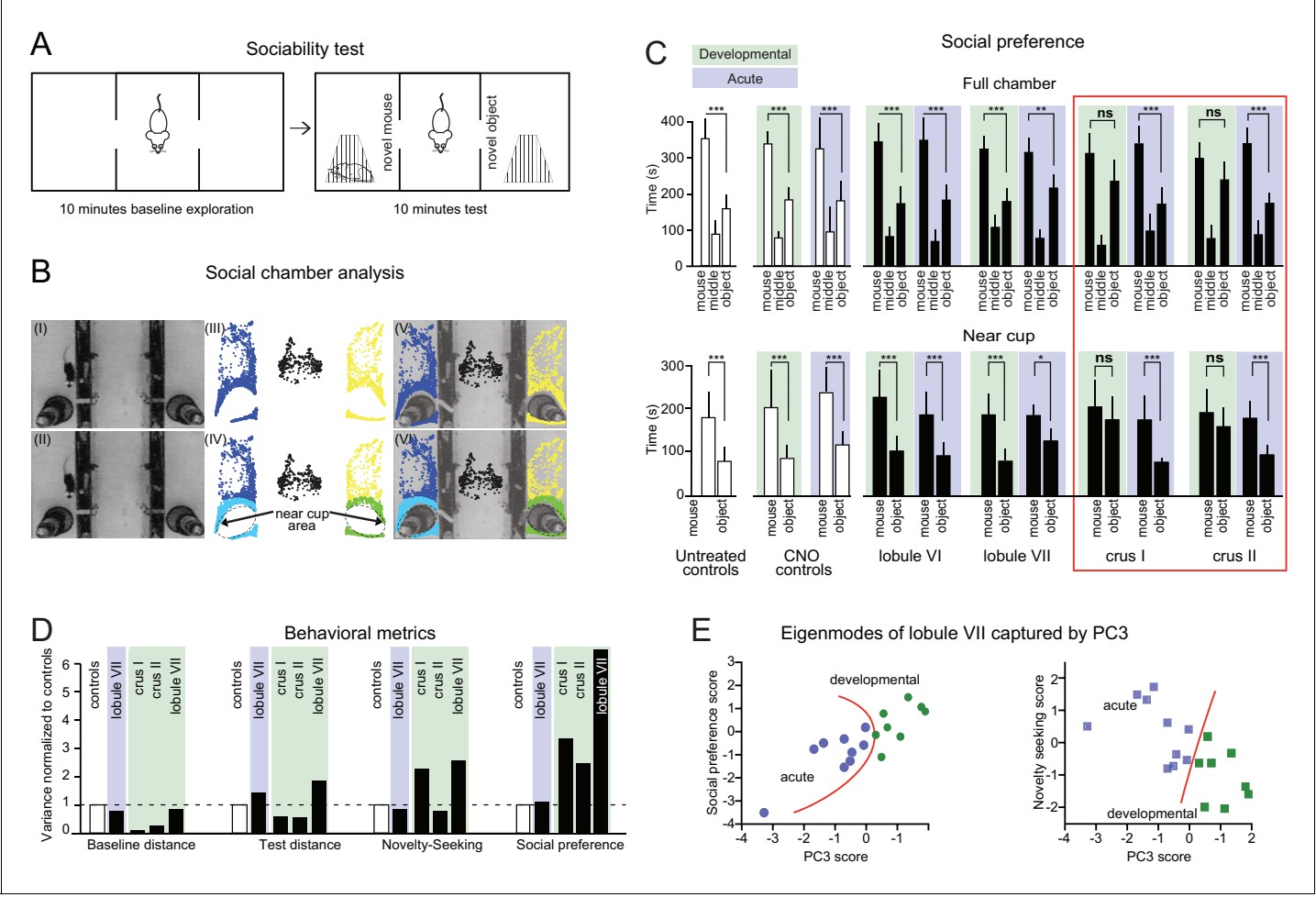

**Figure 6.** Cerebellar developmental impairment in a three-chamber social preference task. (**A**) The three-chamber social preference test. (**B**) Interaction times with under-cup mouse and object were quantified using automated detection of near-cup approaches (see *Figure 6—figure supplement 1*). (**C**) Specific impairments in social preference after developmental perturbation of crus I or crus II but not other lobules, as measured by both time spent anywhere in the chamber and time spent near the cup. Error bars indicate mean ±SD. (**D**) Variance in social behavior metrics points to a stronger effect of cerebellar perturbation on non-motor metrics than on movement. (**E**) Separation of individual mice according to principal component analysis (see *Figure 6—figure supplement 2*). Orange curves indicate a boundary between opposite effects in acutely and developmentally perturbed lobule VII mice. Individual behavioral metrics were insufficient to separate the two groups (see *Figure 6C*). *p<0.05; **p<0.01; ***p<0.001.
DOI: https://doi.org/10.7554/eLife.36401.015

The following figure supplements are available for figure 6:

**Figure supplement 1.** Social chamber metrics.
DOI: https://doi.org/10.7554/eLife.36401.016
**Figure supplement 2.** Social chamber principal component analysis.
DOI: https://doi.org/10.7554/eLife.36401.017

CNO-receiving controls as a baseline. Juvenile activation of DREADDs in lobule VII was associated with prolonged saline-injection-induced grooming bouts (*Figure 8C*). This effect was reversed if the injection contained CNO, suggesting that lobule VII retained some ability to modulate grooming (*Figure 8E*). Thus, normal activity of lobule VII in juvenile life is necessary for regulating several forms of persistive behavior (exploration time and self-grooming) under non-social conditions.

To summarize, at the level of single behavioral metrics we found three developmental deficits: lobule VI was necessary for choice reversal (Y-maze), lobule VII for regulating persistive behavior (EPM, self-grooming) or novelty-seeking (three-chamber task), and crus I/II for the ability to express a social preference. These deficits set the stage to explore cerebellar endophenotypes using dimensionality-reduction and regression methods.

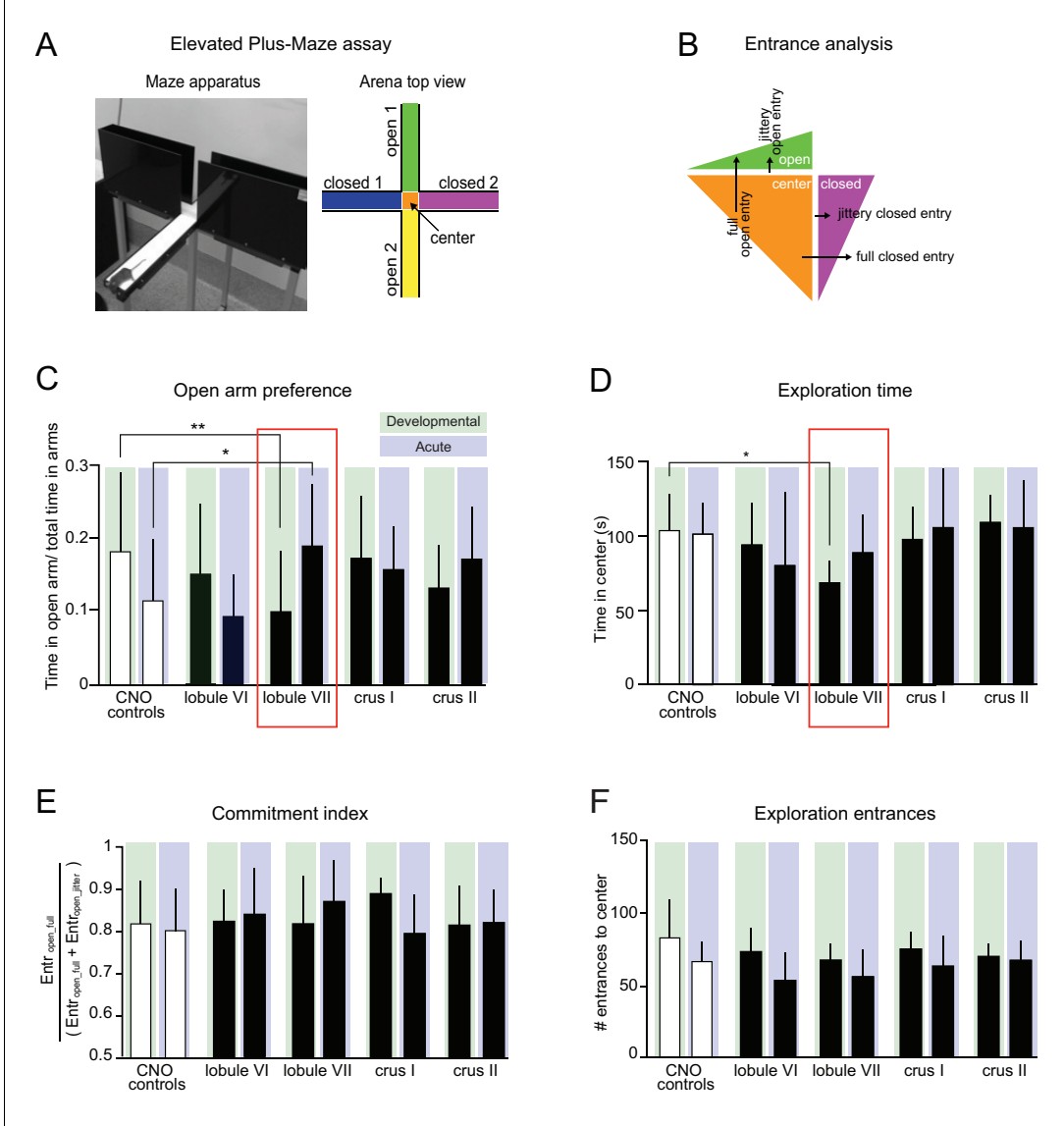

**Figure 7.** Lobule VII-dependent developmental and acute impairment of exploratory behavior. (**A**) Elevated plus-maze (EPM). *Right*, schematic view. (**B**) Entrance analysis. Entrances were defined as either full entries or jittery entries. (**C**) Opposite effects on open-arm preference were found between developmental and acute groups. Developmental perturbation of lobule VII led to reduced open-arm preference. Acute perturbation of lobule VII led to increased open-arm preference (one-way ANOVA, p<0.05, *d* = 0.92, Šidák's multiple comparisons post-hoc test p=0.02). (**D**) Developmental perturbation of lobule VII led to reduced exploration time. (**E**) Analysis of exploration time commitment index (as defined in *Table 2*) found no difference between groups. (**F**) Analysis of exploration entrances in EPM found no difference between groups. Error bars indicate mean ±SD. (*Figure 7—figure supplement 1* and *Figure 7—figure supplement 2*).

DOI: https://doi.org/10.7554/eLife.36401.018

The following figure supplements are available for figure 7:

**Figure supplement 1.** EPM metrics.
DOI: https://doi.org/10.7554/eLife.36401.019
**Figure supplement 2.** Novelty-seeking shows opposite effects of acute and developmental perturbation to lobule VII.
DOI: https://doi.org/10.7554/eLife.36401.020

## Behavioral variation within treatment groups is dominated by nonmotor features

In the past, the tasks of *Figures 4–8* have been used by autism researchers to investigate flexible behavior in mice. However, these tasks also depend on brain mechanisms that are often considered

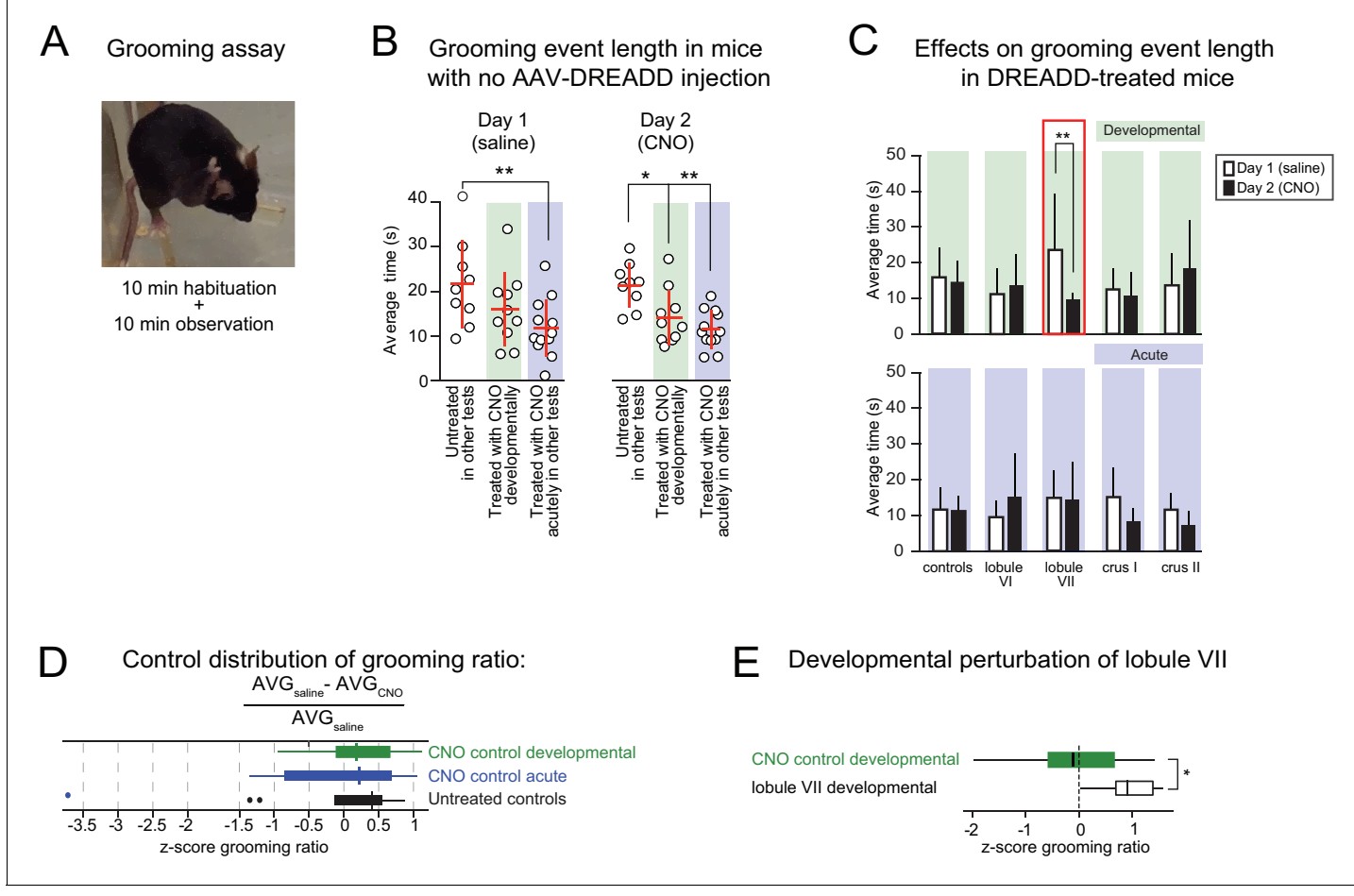

**Figure 8.** Lobule VII-dependent developmental perturbation of grooming. (A) Grooming behavior was manually scored during a 10-min observation window and video-recorded for post-hoc corroboration. A grooming event was defined as a unilateral or bilateral stroke or full-body cleaning (*Kalueff et al., 2016*). Grooming events of less than 1-s duration were excluded from analysis. (B) Average grooming time in response to (*left*) saline injection on day 1 and (*right*) CNO injection on day 2 of testing in untreated mice (no AAV-DREADD injection or CNO) and mice that received CNO developmentally or acutely (color-shaded regions). (C) Injection-triggered grooming in DREADD-treated mice and corresponding controls. Bar graphs show average grooming time in response to the same 2-day injection protocol as in (B). The controls show the same data as the color shaded regions in (B). Only in developmental lobule VII mice was grooming time altered when CNO was injected rather than saline. (D) There were no differences in distributions of the z-score normalized three control groups for grooming ratio defined as the difference between average grooming bout length in 'saline' and CNO conditions relative to 'saline' condition. Outliers defined as points outside the interquartile range (IQR) by more than 1.5 IQR. (E) The change in grooming bout length between CNO and baseline condition was significant only in the developmental perturbation of lobule VII. **p<0.01.
DOI: https://doi.org/10.7554/eLife.36401.021

to be motor in nature. To visualize relationships among task parameters, we used inter-measure correlations. Untreated mice were found to have significant (p<0.01, r-to-z transformation) within-task correlations, often relating to movement (*Figure 9*). Measures for elevated plus-maze were strongly interconnected with one another, suggesting a high degree of redundancy between these measures. More importantly, we were interested to see if these classical tests for autism-like phenotypes were interrelated, to find a commonality between the measures. In untreated and developmentally treated mice, elevated plus-maze measures were also correlated with social preference in the three-chamber test and with grooming ratio, suggesting that some measured features may capture shared capacities that are common to the three tasks.

## Quantifying the dosage-dependence of cerebellar contributions to flexible-behavior endophenotypes

We surmised that cerebellar phenotype intensity might vary as a function of DREADD expression, consistent with the idea of the cerebellum as a feedback controller (*Dean and Porrill, 2014*) and on the approximately-linear transformation of Purkinje cell activity to deep nuclear output (*Turecek et al., 2016*). We used linear regression to quantify the extent to which the spatial reach of expression in the four targeted lobules (lobule VI, lobule VII, crus I, crus II) could account for behavioral metrics. Linear models allowed us to quantify the contribution of each lobule to each measured behavioral parameter.

In our linear models, the extent of DREADD expression in MLIs in each lobule was used as an input variable. To quantify expression, we reconstructed mCherry co-expression from two-photon tomographic images (*Figure 10A* and *Figure 10—figure supplement 1*) or from serial sections (*Video 5*) for all DREADD-injected mice. The extent of expression in each lobule was defined as the fraction of voxels in that lobule in which mCherry was detected. Visualized MLIs appeared dense within labeled volumes, consistent with near-complete efficiency of expression (see *Video 1*). Injections filled 21 ± 12% (average ±SD) of the targeted lobule, and partially spilled into 11 ± 11% of the next-highest expressing lobule.

For each behavioral metric, a linear model was fitted using four regressors, corresponding to the fraction of voxels containing label in each of the targeted lobules (crus I, crus II, lobule VI and lobule VII; *Figure 10B*). Two models were fitted, one for the juvenile-perturbed mice and one for the adult-perturbed mice. DREADD-untreated mice were not included. The weights of each best-fit model (*Figure 10C*) can be interpreted as the influence of expression by that lobule over the behavior of interest.

Weights often had the same sign as the group-level effects shown in *Table 3*, consistent with the hypothesis that the effects were dependent on the dosage of inactivation of the targeted lobules. In juvenile-perturbed mice, weights of greater than 1.75 standard errors were found for crus I on social-chamber performance, as well as for lobule VI in several measures of Y-maze reversal, consistent with the analysis of single-trait and principal component analysis. Thus, crus I and lobule VI appear to have quantitative and specific effects on developmentally acquired social preference and cognitive flexibility. High weights were also found in adult-perturbed mice for Y-maze reversal (crus I/II and lobule VI) and grooming (lobule VI), and to a considerably lesser extent for social-chamber performance (crus I).

Mismatch between linear-model weights and group-level effects could occur for several reasons. For example, a lobule could have multiple conflicting and/or nonlinear effects on behavior. Another possibility is that a group for a single lobule might be statistically underpowered, but dependence emerges when other groups with expression in the same lobule are included. Indeed, acutely injected mice showed strong dependence of grooming on the volume of lobule VI expressing DREADD (2.9 standard errors, *Figure 10C*), despite the fact that the lobule VI group analyzed alone did not show a statistically significant effect on grooming (*Figure 8C*).

## Lobule VI and crus I communicate with neocortical regions that support flexible and social behavior

We next sought to identify distal targets of lobule VI and crus I in the neocortex, which the cerebellum influences via ascending disynaptic pathways (*Strick et al., 2009*). We injected lobule VI and crus I with the anterograde transsynaptic tracing virus HSV-H129 recombinant 772 (*Wojaczynski et al., 2015*), which drives expression of EGFP. We then waited 60 or 80 hr before sacrifice, enough time for viral spread through deep nuclei and thalamus/midbrain to reach neocortex (*Figure 11A*). We counted sections with GFP-expressing neurons in coronal sections spanning a range of neocortical structures and found the strongest expression in motor, somatosensory cortex and taenia tecta (*Table 4*, *Figure 11—figure supplement 1*).

We defined a neocortical region's expression as the number of sections expressing GFP (*Figure 11B*) divided by the number of GFP-positive sections in motor cortex, which was always labeled (*Figure 11—figure supplement 1*). Injection of lobule VI (three mice) led to expression in orbitofrontal, prelimbic, anterior cingulate, and infralimbic cortex, consistent with human mapping (*Buckner et al., 2011*). Injection of crus I (four mice) led to expression in anterior cingulate,

**Table 3.** Cohen's d effect size of motor and non-motor metrics.

**Flexible/Social/Learning metrics**

| | | Y-maze reversal | | Three-chamber | | Elevated plus-maze | | Grooming | Eyeblink conditioning |
|---|---|---|---|---|---|---|---|---|---|
| | | Learning (Final learning) | Persistive behavior (Multisession reversal[1]) | Social preference | Novelty-seeking | Open-arm preference | Exploratory behavior | Grooming ratio | % Conditional responses session 11 |
| Lobule VI | Developmental | 0.1 | *1.1* | −0.1 | 0.7 | 0.3 | 0.4 | −0.6 | −0.3 |
| | Acute | *0.5* | *0.9* | 0.2 | 0.0 | 0.3 | 0.6 | 0.1 | 0.1 |
| Lobule VII | Developmental | 0.6 | −0.3 | −0.3 | *1.1* | *0.9* | *1.7* | *1.2* | nd |
| | Acute | −0.6 | 0.3 | *0.9* | −0.4 | *−0.9* | 0.5 | 0.2 | nd |
| Crus I | Developmental | 0.4 | 0.6 | *1.2* | *−0.8* | 0.1 | 0.3 | 0.0 | *1.6* |
| | Acute | 0.0 | 0.3 | 0.0 | 0.3 | −0.6 | −0.1 | *0.9* | *1.0* |
| Crus II | Developmental | 0.1 | −0.5 | *1.2* | 0.0 | 0.6 | −0.3 | −0.5 | −0.5 |
| | Acute | −0.6 | −0.6 | 0.0 | 0.2 | *−0.8* | −0.2 | *0.9* | 0.3 |

**Movement metrics**

| | | Gait | Y-maze reversal | Three-chamber | EPM |
|---|---|---|---|---|---|
| | | Stance | Distance | Distance Baseline | Distance |
| Lobule VI | Developmental | 0.1 | −0.1 | 0.1 | 0.7 |
| | Acute | −0.3 | **−0.7** | −0.4 | 0.7 |
| Lobule VII | Developmental | 0.2 | −0.2 | −0.3 | *1.2* |
| | Acute | 0.5 | 0.1 | −0.1 | 0.3 |
| Crus I | Developmental | −0.3 | −0.4 | *−1.1* | 0.7 |
| | Acute | **0.7** | 0.2 | 0.2 | 0.0 |
| Crus II | Developmental | 0.1 | −0.3 | −0.2 | *1.2* |
| | Acute | 0.3 | 0.2 | −0.2 | −0.3 |

Effect size, calculated in units (Cohen, 1988) of the two-sample pooled standard deviation, of perturbations on key behavioral parameters. Values in *Bold* indicate statistical significance. Colored fields indicate large effect sizes (d≥0.8); Blue for improvement of function and red for impairment as defined in Table 2.

DOI: https://doi.org/10.7554/eLife.36401.023

The following source data is available for Table 3:

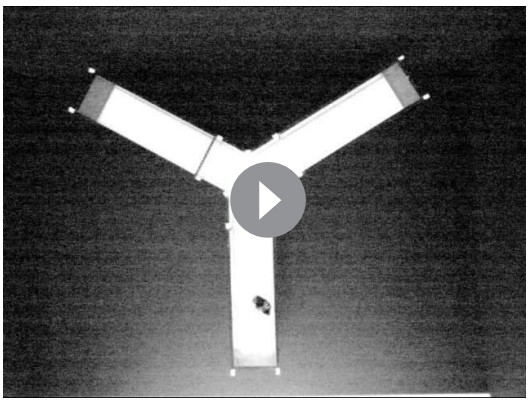

**Video 4.** Example of Y-maze reversal impairment.
DOI: https://doi.org/10.7554/eLife.36401.025

prelimbic, and infralimbic cortex, again consistent with human mapping. Lobule VI was distinguished from crus I by its relatively strong projections to prelimbic and orbitofrontal cortex, regions that play key roles in reward expectation and value-based decision-making (*Rolls and Grabenhorst, 2008*), and would therefore be expected to be specifically regulate reversal learning.

Crus I showed relatively strong projections to anterior cingulate cortex, which participates in flexible and affective cognition (*Apps et al., 2016*; *Devinsky et al., 1995*). A second differentially-strong target of crus I was somatosensory cortex; granule cells in crus I have been reported to respond to orofacial stimuli (*Giovannucci et al., 2017*; *Shambes et al., 1978*), suggesting that this sensory information might be of use in early life to permit - or even drive - the emergence of the capacity for social

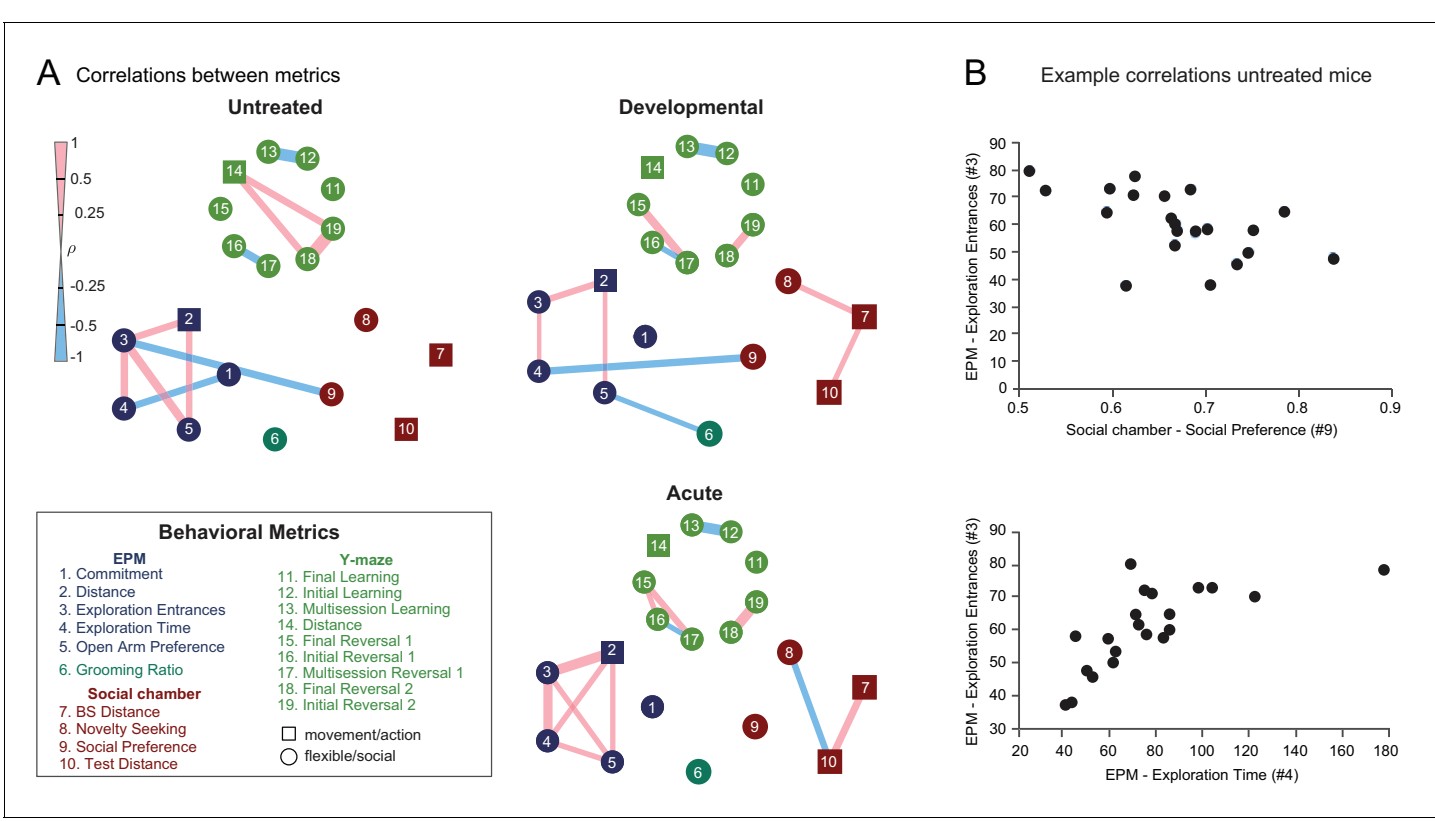

**Figure 9.** Developmental and acute perturbations induce new correlation structure between behavioral metrics. (A) Mouse-by-mouse correlations between pairs of behavioral measures in control mice, developmentally DREADD-activated, and acutely DREADD-activated mice. Significant correlations (p<0.01, t-test) between individual behavioral metrics are indicated by colored bands whose thickness corresponds to Spearman's ρ. Chemogenetic perturbation induced within-task and between-task correlations not seen in untreated mice. (B) Scatter plots showing example relationships between pairs of behavioral measures.
DOI: https://doi.org/10.7554/eLife.36401.026

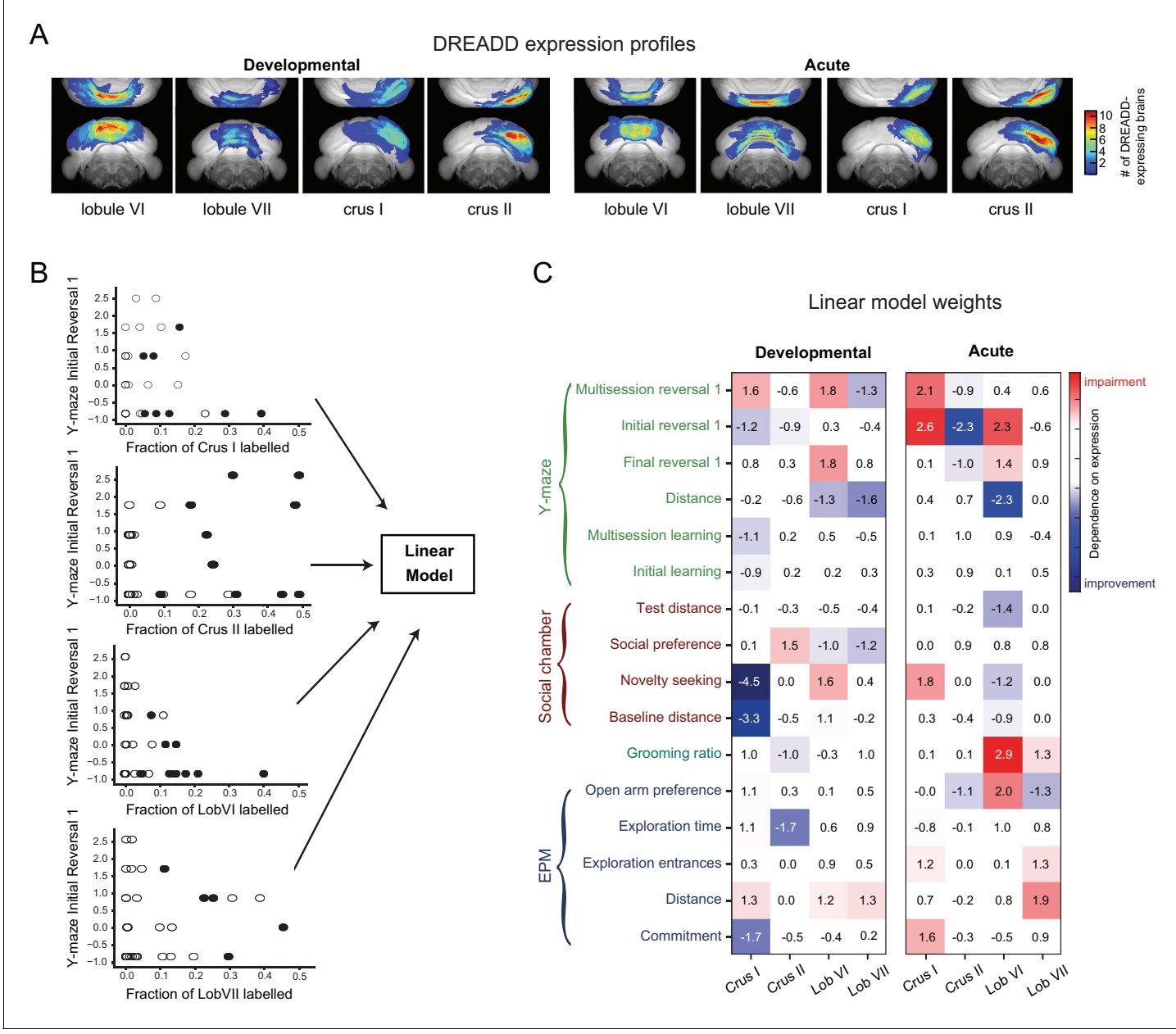

**Figure 10.** Quantitative contributions of lobules to behavioral metrics revealed using a linear model. (**A**) Whole-brain reconstructions of recovered DREADD-mCherry expression. mCherry-positive voxels were recovered from reconstructions from serial two-photon tomographic images of developmentally and acutely perturbed lobule VI, lobule VII, crus I, and crus II mice (*Figure 10—figure supplement 1* and *Video 5*). The color scale indicates how many brains showed expression at a particular location. (**B**) Linear models were used to evaluate the influence of fraction-of-lobule DREADD expression on each behavioral metric. Scatter plots demonstrate example relationships between fraction-of-lobule expression and individual behavioral metrics. Each dot represents one animal. Filled circles represent mice in which the majority of DREADD expression was found in the indicated lobule. (**C**) Regression weights of the best-fit model for each behavioral metric, normalized by the standard error of the weight estimate.
DOI: https://doi.org/10.7554/eLife.36401.027

The following figure supplement is available for figure 10:

**Figure supplement 1.** Whole-brain reconstructions of recovered DREADD-mCherry expression.
DOI: https://doi.org/10.7554/eLife.36401.028

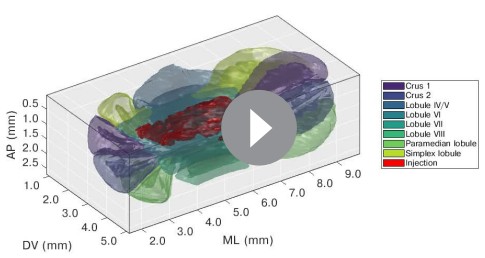

**Video 5.** Registration of the reconstructed cerebellum to the Allen Brain Atlas space. Regions of interest were manually traced in from serial section two-photon images to yield volumetric reconstructions of both anatomical subdivisions as well as the injection spread. After registration to the Allen Brain Atlas, coordinates of the traced regions were transformed to the isotropic reference space.

DOI: https://doi.org/10.7554/eLife.36401.029

preference. These pathways suggest that posterior cerebellar lobules might specifically influence distant neocortical regions to shape the behavioral phenotypes that we tested.

## Discussion

Our findings show that the cerebellum exerts substantial influence over the development of social and flexible behavior. These results could be explained if the cerebellum plays a preprocessing role that, over time, guides the long-term maturation of novelty-seeking and flexible cognition. Cerebellar function and structure are aberrant in the majority of people with autism (*Wang et al., 2014*), a disorder that arises in the first few years of life (*Courchesne et al., 1988*; *Kates et al., 2004*; *Schumann and Nordahl, 2011*; *Wang et al., 2014*). We perturbed in the second month of rodent postnatal life, which approximately corresponds to the first several years of human life as defined by neocortical growth and plasticity (*Bayer et al., 1993*; *Liscovitch and Chechik, 2013*). Many autism susceptibility genes are coexpressed in the cerebellum during postnatal development (*Menashe et al., 2013*;

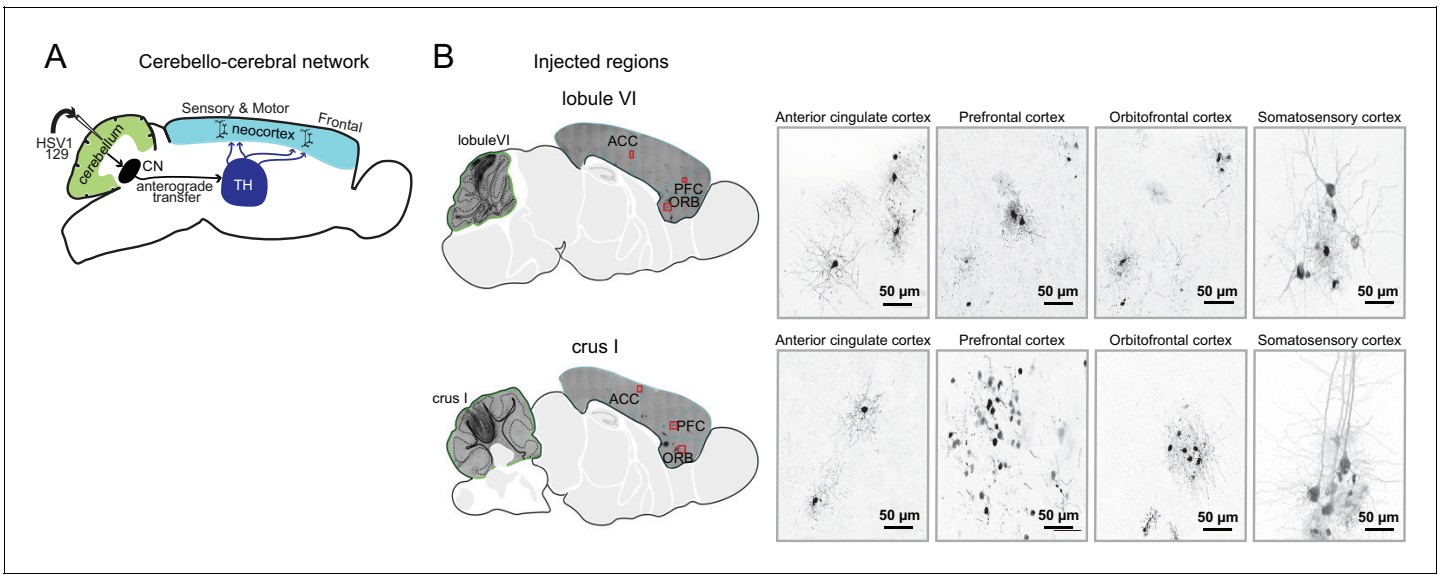

**Figure 11.** Transsynaptic tracing of cerebello-cortical projections. (**A**) Cerebello-cortical pathways. HSV-H129 anterograde tracer injected into lobule VI or crus I reveals specific projections to neocortex (*Figure 11—figure supplement 1*). (**B**) *Left,* Example confocal microscopy images (10x) showing anti-GFP immunoreactivity for HSV-H129 positive cells in *Top,* lobule VI (lateral 0.36 mm) and *Bottom,* crus I (lateral 2.40 mm) to forebrain regions (anterior cingulate cortex, ACC; prefrontal cortex, PFC; orbitofrontal cortex, ORB; somatosensory cortex, SS). *Right,* Example confocal microscopy images (40x) of anti-GFP immunoreactivity for HSV-H129 positive cells in forebrain regions contralateral to the injection site (anterior cingulate cortex, ACC; prefrontal cortex, PFC; orbitofrontal cortex, ORB; somatosensory cortex, SS).

DOI: https://doi.org/10.7554/eLife.36401.030

The following figure supplement is available for figure 11:

**Figure supplement 1.** Transsynaptic tracing of cerebello-cortical projections to motor cortex and taenia tecta.

DOI: https://doi.org/10.7554/eLife.36401.031

**Table 4.** Relative neocortical expression resulting from long-distance tracing.

| | Injection site | |
| | Lobule VI | Crus I |
| Target Region | Relative expression | Relative expression |
| --- | --- | --- |
| Motor cortex | 1.00 | 1.00 |
| Somatosensory | 0.66 | 1.23 |
| Prelimbic | 0.71 | 0.32 |
| Orbitofrontal | 0.72 | 0.22 |
| Anterior cingulate | 0.42 | 0.59 |
| Infralimbic | 0.28 | 0.28 |
| Visual cortex | 0.25 | 0.27 |
| Parietal association | 0.21 | 0.02 |
| Retrosplenial | 0.11 | 0.17 |
| Agranular insular | 0.03 | 0.00 |

DOI: https://doi.org/10.7554/eLife.36401.032

Wang et al., 2014; Willsey et al., 2013) and are required for the normal expression of cerebellum-dependent associative learning (Kloth et al., 2015). Our chemogenetic approach provides a means of disrupting cerebellar circuit function independent of specific genes, thereby allowing relatively direct perturbation of activity as well as the exploration of specific sites within the cerebellum.

How might the cerebellum provide guidance to behavioral development? The cerebellum's circuit architecture allows it to carry out certain types of information processing with exceptionally high computational power. Over half of the mammalian brain's neurons are cerebellar granule cells. Granule cells provide a wide range of efference, sensory, and other signals (Giovannucci et al., 2017; Huang et al., 2013; Wagner et al., 2017) for use in driving Purkinje cell output, which in turn guides action on subsecond time scales. The cerebellum may provide continual feedback to shape nonmotor function, while simultaneously receiving both external information and the brain's own efforts to control behavior (Wolpert et al., 1998).

In rodents, juvenile life is a period of behavioral maturation (Spear, 2000) and neocortical dendritic spine plasticity (Alvarez and Sabatini, 2007). Our experiments have identified juvenile life as a period when disruption of cerebellar output is sufficient to alter the adult expression of cognitive and social capacities. Further experiments are necessary to determine the minimum effective duration of cerebellar disruption, to test whether vulnerability is restricted to specific developmental time periods, and to determine if the long-term behavioral consequences are accompanied by functional or structural alterations in distal brain structures.

The observed anatomical localization of phenotypes is consistent with long-distance projection patterns of the posterior cerebellum, as well as with clinical evidence from related regions in human cerebellum. Although lobular boundaries may not necessarily carry the same functional significance between species, they nonetheless can be used to indicate approximate homology in the anteroposterior direction. We found that lobule VI was necessary for the development of flexible learning, as well as retaining this capacity in adult life. In default-mode human brain imaging, lobule VI is co-activated with cingulate cortex, medial prefrontal cortex, middle/inferior frontal gyri, inferior parietal lobe, medial occipital cortex, thalamus, and basal ganglia (Buckner et al., 2011; Kipping et al., 2013), indicating that incoming synaptic activity (Thürling et al., 2015) to lobule VI encompasses a variety of associative structures. Further, it has recently been reported that lobule VI activity in humans is correlated with non-motor functions, specifically working memory and emotion (Guell et al., 2018). In behaving rodents, lobule VI activity is correlated with head posture (Sauerbrei et al., 2015) and self-generated head movements (Dugué et al., 2017), consistent with sensorimotor function but also with correlated, as yet unidentified events (Sauerbrei et al., 2015). Using transsynaptic tracing, we found that lobule VI's principal neocortical targets included routes

by which lobule VI can influence working memory and processing of reward and thereby contribute to reversal learning.

We found that crus I plays a broad role in the development of reversal learning, novelty-seeking, and most prominently, social preference. In the past, crus I has been suggested as a region of special susceptibility in autism, and has recently been found to influence flexible-behavior phenotypes in mice (*Stoodley et al., 2017*). In adult humans, crus I (along with lobules VI and VII) shows activation during tasks that use language, working memory, executive function, and emotion (*Stoodley and Schmahmann, 2009*). How might these capacities be shaped by crus I processing in early life? One answer may be via sensory processing. In anesthetized rats, granule cells in crus I respond to orofacial (*Bosman et al., 2010*; *Shambes et al., 1978*) stimuli. Our results from viral tracing indeed show that crus I projects strongly to both somatosensory and anterior cingulate cortex, regions relevant for social interactions (*Apps et al., 2016*). Efferent pathways from deep nuclei to extracerebellar brain regions are present in neonatal (P0.5) mice (*Fink et al., 2006*) and project to thalamus as early as embryonic day 16 in rats (*Altman and Bayer, 1997*). In postnatal life, this sensory information might be of use in analyzing, distinguishing, and learning from social and non-social cues. Processing of such information in early life may permit - or even drive - the emergence of the capacity for social preference. Our results suggest that failure to make accurate cerebello-cerebral associations during development could lead to lasting functional consequences. In this way, early-life cerebellar dysfunction could be a cause of later structural and functional alterations elsewhere in the brain (*Wang et al., 2014*).

In contrast to a recent report (*Stoodley et al., 2017*), we found that crus I's role in regulating autistic-like phenotypes was largely restricted to postnatal development. In our hands, perturbation in adult life failed to generate defects in flexible or social behavior. The difference in result could have arisen for several reasons. First, our perturbation of molecular layer interneurons would be expected to reduce the output and modulation of total deep-nuclear output, whereas the previous work inhibited Purkinje cells and would putatively have increased deep-nuclear output. Second, the previous work did not quantify the spatial extent of DREADD perturbation, raising the possibility that injections spilled over into adjacent structures such as lobule VI, which we found to regulate reversal learning. Further experiments are necessary to resolve these differences.

The interpretation of DREADD experiments must be tempered by an understanding of certain technical issues. The DREADD agonist CNO has been demonstrated to be converted to its parent compound, clozapine (*Gomez et al., 2017*; *Manvich et al., 2018*). Clozapine crosses the blood-brain barrier and still activates DREADDs, but it may also exert its own pharmacoactive effects (*Gomez et al., 2017*; *Manvich et al., 2018*). Our CNO-only controls, which revealed subtle behavioral effects, were necessary to establish a baseline for comparison. Clozapine-related effects might also modulate the effects of disrupting cerebellar activity. These questions will be clarified in the future with additional experiments using other chemogenetic (*Magnus et al., 2011*) or optogenetic perturbation approaches.

Our results are of potential relevance to the understanding and treatment of autism spectrum disorder. Autism is notably heterogeneous. We could replicate specific endophenotypes characteristic of autism – social indifference and perseveration – by perturbing specific parts of the posterior cerebellum during postnatal development. Because of its capacity to affect cognitive and social development in humans, the cerebellum has been suggested as a potential site for therapeutic intervention (*Wang et al., 2014*). Our observation of spatially distributed contributions by different anatomical subregions suggests the possibility that such future intervention could be shaped to meet individual requirements.

### Data availability statement

Raw data for this study are available from the corresponding author upon request (including behavioral videos and serial two-photon tomographic brain images of each mouse). Code and data is available at https://github.com/wanglabprinceton/behavioral-development. (*WangLabPrinceton, 2018*; copy archived at https://github.com/elifesciences-publications/behavioral-development).

# Materials and methods

## Key resources table

| Reagent type (species) or resource | Designation | Source or reference | Identifiers |
|---|---|---|---|
| Gene | | NA | |
| Strain, strain background | Mouse: C57BL/6J | The Jackson Laboratory, Bar Harbor, ME | Stock#: 00664 \|Black6 https://www.jax.org/strain/000664 |
| Strain, strain background | Mouse: $L7^{Cre}$;$Tsc1^{flox/flox}$ | From *Tsai et al. (2012)*; *Kloth et al., 2015* | |
| Genetic reagent | | NA | |
| Cell line | | NA | |
| Transfected construct | | NA | |
| Biological sample | | NA | |
| Antibody | anti-GFP Chicken | Aves Labs | Cat#GFP-1020; RRID: AB_10000240 |
| Antibody | Goat anti-Chicken IgY (H + L) Secondary Antibody, Alexa Fluor 647 | ThermoFisher | Cat#A-21449; RRID: AB_2535866 |
| Recombinant DNA reagent | AAV8-hSyn-hM4D(Gi)-mCherry | University of North Carolina Vector Core | |
| Recombinant DNA reagent | AAV1.Syn.GCaMP6f.WPRE.SV40 | Penn Vector Core | lot AV-1-PV2822 |
| Recombinant DNA reagent | HSV1 strain H129 | *Wojaczynski et al. (2015)*; DOI: 10.1007/s00429-014-0733-9 | H129 772 |
| Sequence-based reagent | | NA | |
| Peptide, recombinant protein | | NA | |
| Commercial assay or kit | Elevated Plus Maze (EPM) | Noldus | http://www.noldus.com/animal-behavior-research/solutions/research-small-lab-animals/elevated-plus-maze-set |
| Chemical compound, drug | Clozapine-N-oxide (CNO) | SIGMA-ALDRICH | Cat# C0832 |
| Chemical compound, drug | Dako Fluorescence Mounting Medium | Agilent | Cat# S302380-2 |
| Chemical compound, drug | 15% D-mannitol | SIGMA-ALDRICH | Cat# M4125 |
| Chemical compound, drug | DPBS | ThermoFisher | Cat#14190136 |
| Chemical compound, drug | White tempera paint | Artmind, Tempera Paint | Cat#10091773 |
| Chemical compound, drug | Rimadyl [carprofen] | Zoetis, Florham Park, NJ | http://www.zoetisus.com |
| Chemical compound, drug | Cholera toxin subunit B (CT-B), AlexaFluor 488 Conjugate | ThermoFisher | Cat# C34775 |

*Continued on next page*

*Continued*

| Reagent type (species) or resource | Designation | Source or reference | Identifiers |
|---|---|---|---|
| Chemical compound, drug | Ketamine/xylazine | Met-Vet International /Akorn | RXV CIII (3N)/Cat# 59399-111-50 |
| Software, algorithm | Illustrator CS | Adobe | |
| | Excel | Microsoft | |
| | Ethovision ET | Noldus | |
| | MATLAB | MathWorks | |
| | Python 2.7.14 | Python | |
| | ImageJ | NIH | |
| | Neurolucida | (MBF Bioscience, VT, USA) | |
| | Allen Brain Atlas | (*Oh et al., 2014*). | http://www.brain-map.org |
| | MiniAnalysis program | Synaptosoft Inc, Decatur, GA, USA | http://www.synaptosoft.com/ |
| Other | | | |

## Experimental animals

Experimental procedures were approved by the Princeton University Institutional Animal Care and Use Committee and performed in accordance with the animal welfare guidelines of the National Institutes of Health. Mice used in this study were C57BL/6J (referred to in the manuscript as 'wild-type' mice) males ordered through Jackson Laboratory (The Jackson Laboratory, Bar Harbor, ME). All mice had at least 48 hr of acclimation to the holding facility in the Princeton Neuroscience Institute vivarium before experimental procedures were performed. Mice in the acute cohort, four experimental groups for each lobule (see *Figure 1C*, and *Table 1*), and age-matched CNO controls (n = 10), arrived at 5 weeks of age with five littermates per cage; mice in the development cohort, four experimental groups (n = 40), and age-matched CNO controls (n = 10) arrived at 2.5 weeks of age with five littermates per cage.

Additionally, five more control groups were used for various behavioral experiments as described in the Results section, specifically: (a) 16 adult C57BL/6J male mice that received no treatment and no surgery were used as controls for Y-maze reversal, social chamber, grooming, gait and elevated plus-maze; (b) six adult C57BL/6J male mice that received saline injections but no surgery were used as controls for Y-maze reversal, social chamber, grooming, gait and elevated plus-maze; (c) eight adult C57BL/6J male mice that received GCaMP6f injections in lobule VI and CNO injections were used as sham controls for Y-maze reversal; (d) 16 adult C57BL/6J male mice that received headplate surgery were used as controls for eyeblink conditioning; (e) five adult C57BL/6J male mice that received DREADD injections into the eyeblink zone and saline throughout training served as sham controls for eyeblink conditioning. For eyeblink acute experiments we also injected 4 C57BL/6J male mice directly into a zone in paravermal lobule VI previously shown to directly drive the eyeblink conditioned response (*Heiney et al., 2014*; *ten Brinke et al., 2015*). To benchmark gait deficits we used six adult *L7*[Cre];*Tsc1*[flox/flox] male mutant mice (*Tsai et al., 2012*) in which Purkinje cell degeneration causes ataxic gait.

All mice were housed in Optimice cages (Animal Care Systems, Centennial, CO) containing blended bedding (The Andersons, Maumee, OH), paper nesting strips, and one heat-dried virgin pulp cardboard hut (Shepherd Specialty Papers, Milford, NJ). PicoLab Rodent Diet food pellets (Lab-Diet, St. Louis, MO) and drinking water (or CNO water in the developmental groups) were provided ad libitum. Mice were relocated to clean cages with new component materials every two weeks. All mice were group-housed in reverse light cycle to promote maximal performance during behavioral testing, which took time during the day.

## Animal preparation

Surgeries on mice in the acute and developmental cohorts were performed in accordance with previously published procedures (*Giovannucci et al., 2017*; *Kloth et al., 2015*). In short, mice were anesthetized with isoflurane (5% for induction, 1 – 2% in oxygen; 1 L/min) and mounted into a stereotaxic head holder (David Kopf Instruments, Tujunga, CA) with a heating pad under the ventral surface of the mouse. Puralube vet ointment (Pharmaderm Florham Park, NJ) was administered to the eyes to prevent corneal drying. The scalp was shaved, cleaned, and skin incisions were made along the lambdoid suture and extending caudally from lambda to expose occipital muscle and bone. Minimal muscle was removed to expose the occipital bones. This cut was more extensive in the lobule VII group due to the posterior position of this lobule. Osmotic diuretic drug was administered via intraperitoneal (i.p.) injection (15% D-mannitol in DPBS, 0.6 mL injection for adult, 0.3 mL injection for juvenile) 10 min before opening a craniotomy over the targeted lobule with a 0.5 mm micro-drill burr (Fine Science Tools, Foster City, CA). Sections of Surgifoam absorbable gelatin sponges (Ethicon, Somerville, NJ) immersed in saline and/or artificial cerebrospinal fluid (50 mL ACSF and 100 μL 1M CaCl) were used to stop bleeding, hydrate the skull, and cover exposed brain.

Designer Receptor Exclusively Activated by Designer Drugs (DREADD) viral construct AAV8-hSyn-hM4D(Gi)-mCherry (UNC Vector Core, Chapel Hill, NC) was injected using borosilicate glass capillaries with an outer diameter of 1 mm and internal diameter of 0.58 mm (World Precision Instruments, Sarasota, FL). The pipettes were pulled using the Sutter Micropipette Puller (Model P-2000, Sutter Instrument Company) and bevelled at a 45 degree angle to a bubble number of ~5. Proper fluid flow through pipettes was verified before insertion into the brain. To ensure proper viral spread across lobules of interest, and to account for difference in sizes across lobules and ages during injection times in acute vs developmental group) between 250 and 900 nL of DREADD construct in total was injected per mouse, distributed across four specific locations (see *Figure 1*). Specifically: (1) three injections at two separate depths each (650 μm and 180 μm below the dura) were made for lobule VI acute cohort; (2) three injections at one depth (180 μm below the dura) for lobule VII acute cohort; (3) two injections at one depth (180 μm below the dura) for crus I and crus II acute cohorts; (4) two injections at two separate depths each (650 μm and 180 μm below the dura) were made for lobule VI developmental cohort; (5) two injections at one depth (180 μm below the dura) for lobule VII developmental cohort; (6) one injection at one depth (180 μm below the dura) for crus I and crus II developmental cohorts. The craniotomy was sealed with a silicone elastomer adhesive (Kwik-Sil, World Precision Instruments, Sarasota, FL). In the developmental cohort, the skin caudal to lambda was sutured in place to cover the craniotomy and cerebellum in mice. In the acute cohort, mice underwent headplate attachment immediately following the viral injections. Custom-made titanium headplates (*Kloth et al., 2015*; *Ozden et al., 2012*) were attached to the skull between bregma and lambda using a double layer of quick-drying dental cement (Metabond, Parkell, Edgewood, NY). Mice in the developmental cohort underwent the headplate surgery at P56, following 5 weeks of DREADD agonist clozapine-N-oxide treatment (CNO, C0832-5mg, Sigma-Aldrich, St. Louis, MO). All mice received a non-steroidal anti-inflammatory drug administered post-surgery (0.2 mL, 50 mg/mL Rimadyl [carprofen, Zoetis, Florham Park, NJ], s.c.).

## Behavioral assays

*General environment, transportation, and drug delivery.* All behavioral tests in developmental and acute groups took place between PND 57 and PND 126. The order of behavioral testing was consistent for all groups: (1) social chamber, (2) grooming, (3) Y-maze reversal learning, (4) eyeblink conditioning, (5) elevated-plus maze, and (6) gait. After transportation from the holding facility, mice were acclimated to the behavioral testing room for at least 60 min before commencing a behavioral test. Behavioral testing took place between 9AM and 6PM under low white light illumination, red light (for elevated plus-maze), or darkness to reduce stress and not to disrupt the sleep-wake cycle of the tested mice. For the social chamber assay, novel mice were transported on a separate level of the rolling cart from the experimental mice and the cages were placed on separate levels in the behavioral testing room. Animals without a water bottle were given a 2-ounce HydroGel 98% sterile water gel (ClearH2O, Westbrook, ME) to prevent transient dehydration. In the acute experimental cohort, the DREADD agonist clozapine-N-oxide (CNO) was delivered through i.p. injection at a dose of 1 mg/kg 20 min before the beginning of behavioral testing. Developmental inactivation was achieved

through oral CNO delivery in opaque water bottles at a dose of 10 mg/kg. In both cases, CNO was first dissolved in dimethyl sulfoxide which was then diluted to 1.5% concentration in sterile saline for i.p. injections or drinking water for oral administration. When indicated, mice received saline as a control for i.p. injections.

*Eyeblink conditioning.* Eyeblink conditioning in all tested mice was performed between PND 84 and 126. Animals were trained as previously described (*Giovannucci et al., 2017*; *Kloth et al., 2015*). Mice were habituated to a freely rotating treadmill for over 5 days of graded exposure (15, 15, 30, 45 and 60 min), and then trained for one up to 1 hr a day for a period of 11 days. The unconditional and conditional stimulus (US and CS, respectively) were delivered using a custom-built setup (*Figure 2*). The conditional stimulus was a flash of light (blue 470 nm LED, 'light CS'), 500 ms in duration, contralateral to the US. The unconditional stimulus (US) was a periorbital airpuff (30 – 40 psi), 30 ms, co-terminating with the CS, delivered via a blunt needle placed 5 mm from the cornea. The eyelid deflection was detected using a Hall-effect sensor (AA004-00, NVE Corporation, Eden Prairie, MN) that was mounted above the same eye. The eyelid position was measured by linearly converting a change in magnetic field, due to the displacement of a small neodymium magnet (3 mm x 1 mm x 1 mm, chrome, item N50, Supermagnetman, Birmingham, AL) relative to the sensor position, to a change in voltage. The magnet was attached to the lower eyelid with cyanoacrylate glue (Krazy Glue) under isoflurane anesthesia prior to placement in the eyeblink experimental apparatus. If part of the experimental treatment, mice were given i.p. injections while under this brief anesthesia and allowed 20 min to recover prior to testing. On average mice received 220 trials per day (200 CS-US paired trials and 20 CS-only probe trials). Stimuli were randomly presented with proportion 90% CS +US, 10% CS. Additionally, 10 US-only trials were given to determine the full closure of the eye. All data were analyzed offline with a custom MATLAB (Mathworks, Natick) code as previously described (*Kloth et al., 2015*). The response probability (*Heiney et al., 2014*; *Kloth et al., 2015*) was calculated as the fraction of counted CRs to the total number of counted trials. The response during the presentation to the CS was evaluated as a CR if it exceeded 0.15 (15% of the full range of the UR) between 100 ms and 280 ms after the onset of the CS.

*Gait analysis.* A transparent polycarbonate sheet (61 cm x 46 cm) and two wooden blocks (56 cm x 5.5 cm) were positioned to create a walking path (50 cm x 6.5 cm) for gait analysis and this apparatus was elevated to a height of 56 cm above the ground. A lamp (120V, 100W; Electrix, New Haven, CT) illuminated the path from below. Mice were placed at the beginning of the path and gently encouraged to walk toward a cardboard box covering the terminus of the path. Between three and seven trials were conducted for each mouse. Trials were video-recorded from below with an iPhone6s (Apple, Cupertino, USA) at 30 frames per second (1080 p) (*Video 2*). A ruler was included in the field of view. A customized MATLAB graphical user interface (GUI) was used to extract and organize clips of mouse locomotion from gait videos. The GUI enabled systematic first-stage processing of multi-mouse videos into relevant locomotion sequences of continuous steps. These identified walking clips were subsampled and converted to JPEG format to generate sequences of ten frames per second. A custom Fiji script processed these sequences of frames into binary paw-print masks and z-projected each clip into a single image. These images were used to identify the clip with the longest set of sequential steps for each mouse (mean ± SD, 7.7 ± 1.4 steps) in which paw-prints were clearly identifiable. The images composing these clips were then processed using the Manual Tracker Fiji plug-in to record the location of each paw throughout the sequence. Starting from when all four paws were first in view, each paw was tracked throughout the entire video clip marking the moment when the paw touched the plexiglass with each step, then the video clip was restarted for the next three paws. The tracking positions were superimposed on top of each other to make a trail of footprints and saved along with the tracking results (*Figure 3B* and *Video 2*). Using a custom MATLAB script, the average stride length for each paw was calculated as the average distance between successive paw placements from these locations with a video-specific pixel conversion rate obtained through the ruler in each gait video. The average distances between the right and left fore and rear paws were calculated to define fore and rear stance.

*Grooming assay.* Each mouse was transported to a clean, empty cage for the grooming assay. The cage rested on a wooden platform under a camera (PlayStation Eye) used to record the mouse during testing at 50 frames per second, using a custom-written Python 2.7.6 (Anaconda 1.8.0) script and CLEye Driver (*Figure 8*). Each mouse was first transported to the clean, empty cage for a 10-min unrecorded habituation phase. Next, the cage was moved to the wooden platform and centered

under the camera. The lid was removed and a thin sheet of low-density polyethylene (Kirkland Stretch-Tite Plastic Food Wrap) was stretched to cover the full opening of the cage base. A 10-min test phase was video-recorded and manually scored by an experimenter using an online chronometer to record the start and end of every grooming event. A grooming event was defined as a unilateral or bilateral stroke or full-body cleaning (*Kalueff et al., 2016*). The experimenter sat immobile at a distance of about four feet from the cage. Acquired videos were used to corroborate and verify grooming events. Grooming events less than one second in duration were excluded from analysis (adapted from *Kalueff et al., 2016*). The grooming assay was performed for each mouse once for the CNO condition and once for the saline condition. In the CNO condition across all cohorts, each mouse was briefly anesthetized with isoflurane and administered an i.p. injection of CNO 20 min before beginning behavioral testing. In the saline condition across all cohorts, each mouse received a saline injections before commencing the assay with the habituation phase. The start-time and end-time of grooming events were transferred from an online chronometer to Microsoft Excel using the Text Import Wizard. Data was concatenated for mice in the same age cohort. A custom MATLAB analysis script was used to filter out grooming events less than one second in duration and to extract raw parameters from each ten minute session of recorded grooming events for each mouse for both the CNO and the 'saline' conditions: the cumulative time spent grooming during the 10-min session, the number of independent grooming events, and the mean length of time between consecutive grooming events.

*Water Y-maze assay.* A transparent polycarbonate apparatus was constructed in the shape of a Y with symmetrical arms each measuring 33 cm in length from the center of the apparatus and 7.5 cm in width. The sides of the apparatus were 0.5 cm thick and were 20.3 cm in height. Notches were made at a distance of 9.5 cm from the center of the apparatus in each arm to allow insertion of a polycarbonate sheet in the Forced condition. A Pyrex glass container was inverted and used as a submerged platform measuring 5.9 cm in height from the bottom of the apparatus. The apparatus was filled with water to a depth of 10 cm. ACMI certified hypoallergenic non-toxic white paint (Art-Minds, Tempera Paint) was dissolved in the water to eliminate any platform visibility. At the end of the experimental day, all water was removed from the maze and the maze was cleaned with tap water, tissues, 70% ethanol and left to dry overnight. A black poster-board screen bordered three sides of the apparatus. A camera (PlayStation Eye) was mounted above the apparatus to record a field of view containing the entire apparatus at 50 frames/s, using a Python 2.7.6 (Anaconda 1.8.0) script and CLEye Driver (https://codelaboratories.com/products/eye/driver/). The mouse was lifted by the tail and inserted into the water 2 cm in front of the apparatus edge for each swimming trial. The tail was released in synchrony with initiating the video recording.

Testing was performed for five consecutive days: Day 0 - Habituation; Day 1 - Acquisition; Day 2 - Test; Day 3 - Reversal; Day 4 - Reversal (*Figure 4A*). Habituation consisted of three 60 s trials of continuous swimming in the apparatus without a platform, each trial beginning at a different arm of the maze. Between 10 and 30 s of rest were given between trials, depending on the condition of the mouse. Three mice from the lobule VII injection group were removed from testing during habituation due to swimming concerns; injecting this posterior lobule required removal of overlying muscle, potentially resulting in a lack of proper neck mobility.

The Acquisition phase contained four sessions, each with five consecutive trials of 40 s duration each. The Y-maze apparatus contained a platform at the end of either the left or the right arm of the Y-maze. The arm with a platform was randomly chosen for each mouse. In the Test phase, each mouse experienced five trials with the platform in the same location as randomly determined in the Acquisition phase. Mice were required to have at least 80% success in their first choice turn-direction order to proceed with the behavioral assay. All animals included in the analysis passed this criteria and continued with the assay. In the Reversal phases, the location of the platform was the opposite of the trained location: mice trained in Acquisition and Test with the platform at the end of the right arm of the apparatus were introduced to a Y-maze apparatus with the platform at the end of the left arm, and vice versa. Each Reversal phase consisted of four separate sessions of five consecutive trials each of 40 s duration, followed by a fifth forced session in which a polycarbonate sheet was inserted into notched slots to prevent entry into the arm without the platform. After the conclusion of each swimming trial, the mouse was removed from the maze or platform and dried shortly with tissue before the start of the next trial. Between sessions, mice were placed in a clean cage with bedding under a heating lamp (Electrix, New Haven, CT; 120V, 100W) until fully dry, then replaced in their

original cages until beginning the next session. Mice were trained in groups of five to seven, thus maintaining an approximately equal amount of rest time between sessions for all mice. Performance metrics regarding the first-choice turn direction of the mouse in the Y-maze behavioral assay were determined through manual scoring and corroborated through an automated custom Python package. This analysis package tracked the position of the mouse during navigation of the Y-maze to determine first-choice turn direction for each trial as well as distance traveled and average velocity (*Video 6*).

*Social chamber assay.* The social chamber apparatus was custom-constructed according to established specifications (*Nadler et al., 2004*) as a transparent polycarbonate box with outer dimensions of 40.5 × 20×22 cm (width, length, height). A piece of Nalgene Versi-Dry Surface Protector material (Thermo Fisher Scientific, Waltham, MA) was cut to these outer dimensions and inserted to line the bottom of the apparatus for each mouse. Two removable polycarbonate partitions evenly divided this outer box into three chambers as depicted in *Figure 6A*. Each of these partitions contained a sliding polycarbonate sheet on its interior side that could be lifted and latched in place to expose a passageway between chambers measuring 10 cm in width by 5 cm in height. Wire cups (Galaxy Cup, Spectrum Diversified Designs, Inc., Streetsboro, OH) measuring 11 cm in height and 10.5 cm in diameter were used as the novel object and to contain the novel mouse. The 1 cm separation between bars on the cups enabled sharing of olfactory cues, visual cues, and prevented biting. 500 ml Pyrex beakers filled with water were placed on top of each cup to prevent climbing and to secure cup location. Between mice, all surfaces of the apparatus as well as the wire cups were cleaned with 70% ethanol followed by distilled water and left to dry. Black poster-board screens bordered three sides of the apparatus. The remaining side of the apparatus faced a blank wall. A camera (PlayStation Eye) was mounted above the apparatus to record a field of view containing the entire apparatus at 50 frames/s, using a custom-written Python 2.7.6 (Anaconda 1.8.0) script and a CLEye Driver. Testing was performed for 1 day and consisted of three phases (Habituation, Baseline and Test). In the Habituation phase (10 min; not recorded), the test mouse was placed in the middle chamber with the doors to the adjacent chambers closed. The doors were opened to begin the Baseline phase (10 min; recorded) in which the test mouse could explore the full, empty apparatus. At the conclusion of the Baseline phase, the test mouse was guided back to the middle chamber and the doors were closed. The novel mouse was placed inside an inverted cup in the mouse chamber and the novel object, an identical cup, was placed in the object chamber. The doors were opened to begin the Test phase (10 min; recorded) in which the test mouse could explore the full apparatus. Established procedure was followed (*Yang et al., 2011*) for training the novel mice, conducting the experimental sociability assay, and cleaning the apparatus between mice and between testing days. Novel mice were screened for consistent excessive biting of cup wires or scratching of the Nalgene Versi-Dry Surface Protector material (Thermo Fisher Scientific, Waltham, MA) lining the apparatus during two 15-min sessions. Novel mice were rotated throughout each day of behavioral testing such that each mouse had at least 2 hr between uses. Novel mice were used for a maximum of two months, at which time they were sacrificed and new novel mice were trained according to this training paradigm. The assay began with acquiring 185 frames (5 s) of background video containing only the empty behavioral apparatus. After placing the mice in the apparatus, the experimenter was not visible to the test mouse or the novel mouse during any phase of the assay. Social Chamber assay analysis was conducted through processing of the recorded videos using a semi-automated custom Python package through a graphical user interface, coordinates of the chamber corners and the outlines of the novel mouse and object cups were collected. Using a threshold-detection algorithm subtracting a pixel-wise mean of the relevant background frame pixels, the location of the mouse throughout each experimental phase was calculated and stored at a sub-sampled rate (similar tracking software as used in Y-maze, see *Video 6*). Baseline and Test experimental phases were processed

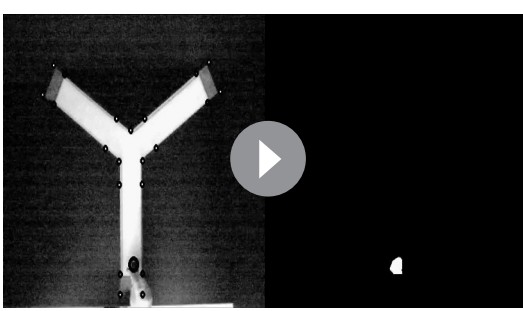

**Video 6.** Example of Y-maze tracking.
DOI: https://doi.org/10.7554/eLife.36401.033

independently. Subsequent group analysis was performed with custom MATLAB scripts.

*Elevated plus-maze assay.* The polycarbonate elevated plus-maze (EPM) consisted of two shielded and two open arms (*Figure 7A*). The base of both arms was constructed with a panel of opaque white polycarbonate embedded in a layer of black polycarbonate. Each full arm measured 77 cm in length and 7.5 cm in width yielding a square center area between the arms of 7.5 cm by 7.5 cm. The sides of the closed arms were constructed of a panel of opaque black polycarbonate 0.3 cm in thickness extending from the juncture of each closed arm with the central area around the full perimeter of each closed arm at a height of 21.5 cm from the closed arm platform. A stabilizing metal frame elevated the plus base to a height of 74 cm above the floor. The illumination source in the room was a single red light. The apparatus was cleaned with 70% ethanol and water and allowed to dry between mice. Animals were brought to the testing room at least 60 min prior to the onset of the experiments to allow habituation. At the onset of the test mice were placed in the central square area between the two arms of the plus maze concurrent with starting the video-tracking software in Ethovision XT (Noldus, Leesburg, VA). Each mouse experienced a single trial consisting of exploration of the full apparatus for 10 min.

Ethovision XT (Noldus, Leesburg, VA) was used to track the centroid of the mouse throughout the ten minute session, with a user-drawn outline of the full arena as well as user-drawn rectangular outlines of both closed arms, both open arms, and the central crossroads area (*Figure 7A–B*). This analysis extracted time spent in each defined area, transitional crossings between each defined area, total distance traveled, and average velocity. Subsequent analysis was performed using MATLAB.

To capture this a variety of capacities, including anxiety and exploratory behaviors (*Holmes et al., 2000*), we quantified the ratio of time spent in open arms to total open-plus-closed-arm time ('open-arm preference'), number of crossings from the center of the maze to either arm ('exploration entrances'), amount of time the mouse spent in the center of the maze ('exploration index') and a ratio of full crossings divided by full-plus-incomplete crossings, where incomplete crossing was defined as stretching of the torso across the entrance border without making a full entrance ('commitment', *Figure 7E* and *Table 2*).

*Social chamber and Y-maze mouse tracking.* Video was acquired at 50 Hz, 640 × 480 pixel resolution, using a Playstation Eye camera with the CLEye Driver. For each behavioral session, mouse tracking was performed using custom software written in Python (WangLab Princeton2018. - https://github.com/wanglabprinceton/behavioral-development) by the following procedure: a background image was computed as the gaussian-filtered mean projection of a 15 to 30 s period prior to insertion of the mouse into the apparatus. For each frame in the experimental session, the difference between the gaussian-filtered frame and the background image was computed. The difference image was thresholded according to a user-defined value, and edges of this binary image were detected using the Canny algorithm. Contours were extracted from the edge image, and the contour with the largest area was taken to represent the approximate outline of the animal. Contours were excluded if their center was greater than 100 pixels away from the location in the previous frame, if they were located outside the user-defined bounds of the apparatus, or if they were inside other user-defined areas containing features that interfered with tracking accuracy. For frames in which the algorithm found no suitable contour, the position of the mouse was inferred to be in its most recent location. For the social chamber the boundaries of the three chambers were demarcated manually and the time spent in each chamber was determined from the number of frames in which the animal was detected in each chamber. For Y-maze, the location of the mouse in each frame was automatically assigned to a specific sub-portion of the maze using fiducial markers on the maze apparatus. Code for tracking is available from the corresponding author upon request.

## Tissue processing and histological procedures

After successful completion of all behavioral assays, experimental mice were anesthetized with an overdose of ketamine (400 mg/kg)/xylazine (50 mg/kg) (i.p.) and perfused with 4% paraformaldehyde (PFA). Brains were stored in 4% PFA for three to six hours at room temperature followed by overnight incubation in 4°C. After rinsing with phosphate buffered saline (PBS) the brains were placed in PBS with 0.1% sodium azide and shipped to TissueVision (TissueVision, Somerville, MA) for processing via serial two-photon tomography imaging. Endogenous tissue autofluorescence (500 – 540 nm) and mCherry DREADD signal (600 – 680 nm) were imaged in coronal slices at intervals of 50 µm. Images received from TissueVision were registered to the Allen Brain Mouse 25 µm slice

Common Coordinate Framework Reference Atlas Version 3 (*Lein et al., 2007*); see 'Quantification of the injection spread' for details.

Mice used for characterization of the DREADD expression pattern were anesthetized with an overdose of ketamine (400 mg/kg)/xylazine (50 mg/kg) (i.p.) and perfused with 4% PFA 4 to 5 weeks following the injection (N = 5 C57BL/6J males injected with the AAV8-hSyn-hM4D(Gi)-mCherry virus; viral injections and surgeries were performed as described above). Following 2 hr post-fixation at room temperature, cerebella were isolated from the rest of the brain, incubated overnight at 4°C in 10% sucrose and embedded in gelatin (12% gelatin/10% sucrose). Gelatin embedded brains were then hardened for 1.5 hr at room temperature in 30% sucrose/10% formaldehyde, incubated overnight at 4°C in 30% sucrose, rapidly frozen, sectioned sagittally at 50 µm and collected in 0.1 M PBS. Sections were processed for immunohistology by washing with PBS and counterstained with DAPI for 10 min (30 µl/ml) and mounted on glass slides with Vectashield anti-fade mounting medium (H-1000; Vector laboratories, USA). Brains from four mice used for in vivo electrophysiology were isolated and processed in the exact same way.

Brains from mice injected with the herpes simplex virus (HSV) one strain H129 causing neuronal expression of EGFP (eight mice; four lobule VI injections and four crus I injections; see 'Long distance tracing' for details) were first anesthetized with isoflurane in their home cages before being injected with ketamine (400 mg/kg)/xylazine (50 mg/kg) (i.p.) and perfused under the animal biosafety level 2 conditions with 4% paraformaldehyde with heparin (20 units/ml) were isolated, postfixed, incubated and prepared for sectioning as described above. Whole brain sagittal sections were cut at 50 µm and collected in 0.1 M PBS. Sections were processed for immunohistology by washing with PBS and incubating for 1 hr at room temperature in a blocking buffer (10% normal goat serum, 0.5% Triton in PBS) prior to a 3-day incubation at 4°C in PBS buffer containing 2% NGS, 0.4% Triton and the chicken anti-GFP primary antibody (GFP-1020, Aves Labs Inc., Oregon, USA; 1:1000, previously described in *François et al., 2017*). Sections were subsequently washed in PBS, incubated for 2 hr at room temperature in the PBS buffer with goat anti-chicken Alexa Fluor 647-conjugated secondary antibody (A-21449; Thermo Fisher Scientific, MA, USA, Invitrogen; 1:200), mounted on glass slides and covered with Vectashield.

Acute 250 µm sagittal cerebellar slices from two C57BL/6J males injected with AAV8-hSyn-hM4D (Gi)-mCherry virus used for in vitro electrophysiological experiments were transferred from the artificial CSF (aCSF) to 4% paraformaldehyde and stored overnight at 4% paraformaldehyde. Fixed slices were then washed with PBS and mounted on glass slides with Vectashield.

The majority of brains were imaged via serial two-photon tomography by the TissueVision company. A smaller subset of brains were cut with a cryostat and imaged at 10x magnification on an epifluorescence Nikon Eclipse Ti microscope (Nikon Instruments Inc, Tokyo, Japan) or on an epifluorescence Zeiss Axioplan2 upright fluorescence microscope (Carl Zeiss Microscopy, Germany). In order to obtain high-resolution images, selected sections were scanned with a Leica SP8 confocal laser-scanning microscope (Leica Microsystems, Germany) using 10x, 40x or 63x objectives, hybrid (HyD) detectors for sensitive detection, and sequential scan mode. Images used in *Video 1* were obtained at 40x by two-photon imaging using a 760 nm wavelength from a 50 µm sagittal cerebellar section.

## Registration of the injection signal

In order to standardize histological expression analysis across animals, we devised a pipeline for sample registration to the Allen Brain Atlas (*Oh et al., 2014*). We opted for a structure-guided registration procedure based on landmark registration due to serial two-photon imaging issues including brain-to-brain variability of fluorophore expression intensity, missing sections, tissue tearing, and deformation.

First, two-dimensional boundaries of anatomical regions of interest within the cerebellum were manually traced within each section for each sample as well as the reference volume (Allen Brain Atlas v3, 25-micron voxels) using Neurolucida software (MBF Bioscience, VT). Additional tracing was performed to outline the injection volume within the same images for the sample brains. Once traced, contours were extracted as a set of line segments defining closed polygons for each connected component within a section. The interior of section-wise polygons were binarized and the volume was resampled to yield an isotropic volume for each structure. Binary volumes were then

converted to triangular surface meshes via isocontour thresholding at isovalues of 1, generating a set of vertices and edges reflecting the underlying closed volume to be used for registration.

In order to use the shape and location of brain structures that were clearly visible as landmarks for aligning the sample to the reference volume we devised a point-based registration procedure. For a given pair of surface meshes, we first applied a rigid registration between the two sets of vertices to approximately account for translation and orientation differences. Then, this registration was refined through a consensus-based nonlinear point set registration algorithm (*Myronenko and Song, 2010*). As the number of vertices are preserved through this approach, edges between them that define the structure surfaces were also unaltered, allowing us to then reconstruct each sample's closed structures in the registered (reference) coordinates. The transformation obtained from this registration was then applied to each Region of Interest (ROI) and voxel coordinate in the sample volume to align all of the structures (including injection signal) to the reference atlas. Registered surface meshes were then binarized by testing each voxel for inclusion in each closed surface, yielding binary volumes of the same shape, resolution and coordinates as the reference volume for subsequent analyses (Video 5). DREADD mice were classified according to whichever lobule showed the largest fraction to express fluorescence. In nearly all cases, this was the same as the lobule that was originally targeted for viral injection.

For joint behavioral-anatomical analysis, viral expression volumes were max projected across the anterior-posterior axis to yield mediolateral/dorsoventral projections, and then further reduced to vectors by taking the average number of expression signal voxels along each coordinate of the mediolateral axis. In 'masked' analyses, the ROI under consideration was used as a voxelwise inclusion filter before reducing sample volumes to expression vectors. All correlations were computed using Spearman's $\rho$ and p-values using exact permutation distributions as implemented in MATLAB's corr function (MathWorks Inc.).

For the linear model analysis, models were fit relating anatomical expression to each behavioral metric with continuous values (all but Y-maze Final Learning, Initial Reversal 2 and Final Reversal 2). Given that the control groups had no expression only experimental (injected) groups were used. Therefore, we regressed the behavioral scores of adult and juvenile experimental cohorts with DREADD injections, onto their respective expression profiles. One model was fit for each behavioral metric, with the value of that metric for each mouse as the predicted variable. Inputs to the model consisted of four values for each mouse: the fraction of lobule containing label for each of lobule VI, lobule VII, crus I, and crus II. Coefficients were fit for each of these terms along with an intercept term. The fraction of expressing voxels for each mouse is shown in *Figure 10—figure supplement 1*. Contribution of anatomical expression in each lobule to each behavior was quantified as the best-fit coefficient of the model fit, normalized by the coefficient standard error. Coefficients and standard errors were obtained by fitting the linear models using the statsmodels 0.8.0 package in Python 3.6.2.

## Quantification of behavioral endophenotypes

As a means of describing the natural underlying structure of the behavioral phenotypes measured through the assay-specific metrics, we performed principal component analysis (PCA) on the set of observations in control mice to define a linear subspace of correlated behavioral measures. The resulting principal components or 'eigenbehaviors' (linear combinations of behavioral metrics) were used as a basis onto which both control and experimental mouse data were projected after normalization by centering to the mean and standard deviation of the controls. Both principal component scores and normalized raw metrics (z-scored metrics described in *Table 2*) were compared between experimental groups and controls via standard nonparametric distribution test, the two-sample Kolmogorov-Smirnov test, as implemented in MATLAB's kstest2 function (MathWorks, Inc.). High-contribution PCs were defined as those for which the experimental group differed from controls in their means by at least 0.01, accounted for at least 10% of the total behavioral variance of the experimental group for that task, and reached $p < 0.05$ two-tailed significance by a two-sample t-test.

To construct the correlation graphs across all assays (*Figure 9*), pairwise correlations were computed between control-normalized metrics of sample groups. Correlation values were computed as Spearman's rank correlation coefficient ($\rho$) and p-values were determined using exact permutation distributions as implemented in MATLAB's corr function (MathWorks Inc.).

## Acute brain slice experiments

Two male, 6-week-old C5B7L/J6 mice were anesthetized with isoflurane and injected in lobule VIb with AAV8-hSyn-hM4D(Gi)-mCherry virus as described in 'Animal Preparation'. Two to 5 weeks later, cerebella were removed and dissected in ice-cold oxygenated slicing medium containing (in mM): 93 N-methyl-D-glucamine HCl, 2.5 KCl, five sodium ascorbate, two thiourea, three sodium pyruvate, 0.5 $CaCl_2$, 10 $MgCl_2$, 30 $NaHCO_3$, 1.2 $NaH_2PO_4$, 20 HEPES, and 25 D-glucose (mOsm 300, pH 7.4). 250 µm sagittal slices of the vermis were cut on a vibratome (VT2000S, Leica, Germany). Slices were incubated in slicing medium at near physiological temperature (34°C) for 2 mins and transferred to 34°C aCSF for 30 min. Slices were then held at room temperature in a chamber filled with oxygenated aCSF covered with aluminum foil to prevent too much light exposure and used within 6 hr. Experiments were performed in aCSF at near physiological temperature ~33°C. The patch pipettes (6 – 8 MΩ) were filled with intracellular solution containing (in mM): 120 potassium gluconate, 9 KCl, 10 KOH, 3.48 $MgCl_2$, 4 NaCl, 10 HEPES, 4 $Na_2ATP$, 0.4 $Na_3GTP$, 17.5 sucrose and 10 µm Alexa 488 (at pH 7.25). CNO (10 µM in aCSF) was applied to the slices using positive pressure via another patch pipette (>2 MΩ) positioned above the recording area. Data was stored and collected for offline analysis using the MiniAnalysis program (Synaptosoft Inc., Decatur, GA; http://www.synaptosoft.com/; n = 7 cells).

## In vivo electrophysiology

Four male, 6-week-old C5B7L/J6 mice were anesthetized with isoflurane and injected in lobule VI with AAV8-hSyn-hM4D(Gi)-mCherry virus as described in the 'Animal Preparation' section. Following the injection mice were fitted with a cranial 3-mm-wide window covered by a removable silicone plug (Kwik-Sil, WPI) and a custom-made two-piece headplate (*Figure 1E* bottom; *Giovannucci et al., 2017*). Two weeks later, mice were lightly anesthetized with isoflurane, head-fixed and placed on a freely rotating disk. The DREADD mCherry expression was confirmed by placing mice the mice under an epifluorescence Leica MZ16FA microscope. The day following this confirmation the mice were again lightly anesthetized with isoflurane, headfixed and placed on a freely rotating treadmill inside a Faraday cage. The silicone plug covering the DREADD-expressing part of the cerebellum was removed and the exposed brain was covered with sterile saline. Single-unit extracellular signals were recorded in awake mice using borosilicate glass electrodes filled with 2 M NaCl. Purkinje cells were identified by the presence of complex and simple spikes. Single units were then confirmed during offline analysis by the characteristic short pause in simple-spike firing that follows each complex spike (*Badura et al., 2013*). Purkinje cells were recorded in baseline condition and upon topical application of CNO (10 µM in sterile saline). The recording site was confirmed by injecting ~100 nl of 1% cholera toxin subunit B (CT-B), Alexa Fluor 488 Conjugate (C34775, ThermoFisher Scientific) dissolved in sterile saline (0.9% NaCl) (*Figure 1E* top). Immediately after acquiring the post-CNO-application recordings and injecting the cerebella with CT-B, mice were injected with an overdose of ketamine (400 mg/kg)/xylazine (50 mg/kg) (i.p.) and perfused with 4% paraformaldehyde to preserve the tissue for histological verification of the injections. Data was stored and collected for offline analysis using SpikeTrain (Neurasmus B.V., The Netherlands, www.neurasmus.com), running under MATLAB (Mathworks, MA).

## Long-distance tracing

The HSV1 strain H129, which contains the EGFP transgene, was used for multisynaptic anterograde tracing (H129 772, *Wojaczynski et al., 2015*); aliquots grown/titered 1/21/15 at $9.02*10^8$ pfu/ml before single freeze-thaw. Eight adult C57BL/6J males were injected with the HSV1 strain H129 in the Biosafety Level 2 (BSL2) facility (N = 8 mice; four lobule VI injections and four right crus I injections; see 'Animal Preparation' for details regarding surgery). Virus was diluted 10x in sterile saline before injection. Following surgery mice were kept in the BSL2 facility. After 60 or 80 hr post-op mice were first anesthetized with isoflurane in their home cages and then injected with ketamine (400 mg/kg)/xylazine (50 mg/kg) (i.p.) and perfused under Animal BSL2 conditions with 4% PFA with heparin (20 units/ml). The number of sections per region (motor cortex, somatosensory, prelimbic, orbitofrontal, anterior cingulate, infralimbic, visual cortex, parietal association, retrosplenial, and agranular insular) with GFP was counted. The relative neocortical expression was calculated by dividing each region's count by the number seen in motor cortex.

## Statistics

Statistics and grouping were performed using MATLAB, Python, or GraphPad Prism. Unless stated otherwise, data are presented as mean ± SD. Two statistical assays were employed to determine differences in behavioral measures resulting from lobule and temporal specific perturbations: group mean comparison through ANOVA with Šidák's multiple comparisons post-hoc test, and a distribution comparison using the two-sample Kolmogorov-Smirnov (KS) test. Inclusion of the non-parametric KS enabled identification of potential differences in the overall distribution of behaviors in addition to distribution mean and median.

## Acknowledgements

We thank Chris De Zeeuw and the laboratory of SW for support and commentary during this project, Chris De Zeeuw, Kelly Seagraves, Junuk Lee, and Lindsay Willmore for reading the manuscript, Julia Epelbaum, Zhenyu Gao, and Laura Lynch for help with experiments, and Halina Goraczniak and Lynn Enquist for HSV-H129 virus. This work was supported by Innovational Research Incentives Scheme VENI (NWO, ZonMw) (AB), the Nancy Lurie Marks Family Foundation, NIH R01 NS045193 and R01 MH115750 (SW), the New Jersey Commission on Brain Injury Research CBIR16FEL010 (JV), National Science Foundation Graduate Research Fellowship DGE-1148900 (TDP), NIH F31 NS089303 (TJP), NIH F30 MH115577 (BD), and the Rutgers Robert Wood Johnson Medical School-Princeton University M.D. - Ph.D. Program (BD and TJP).

## Additional information

### Funding

| Funder | Grant reference number | Author |
|---|---|---|
| Nederlandse Organisatie voor Wetenschappelijk Onderzoek | Innovational Research Incentives Scheme VENI (NWO,ZonMw) | Aleksandra Badura |
| Nancy Lurie Marks Family Foundation | | Samuel S-H Wang |
| National Institutes of Health | R01 NS045193 | Samuel S-H Wang |
| New Jersey Commission on Brain Injury Research | CBIR16FEL010 | Jessica L Verpeut |
| National Science Foundation | Graduate Research Fellowship DGE-1148900 | Talmo D Pereira |
| Rutgers Robert Wood Johnson Medical School-Princeton University M.D.-Ph.D. Program | | Thomas J Pisano Ben Deverett |
| National Institutes of Health | R01 MH115750 | Samuel S-H Wang |
| National Institutes of Health | F30 MH115577 | Ben Deverett |
| National Institutes of Health | F31 NS089303 | Thomas J Pisano |

The funders had no role in study design, data collection and interpretation, or the decision to submit the work for publication.

### Author contributions

Aleksandra Badura, Conceptualization, Resources, Data curation, Supervision, Funding acquisition, Investigation, Writing—original draft, Project administration, Writing—review and editing; Jessica L Verpeut, Conceptualization, Data curation, Formal analysis, Funding acquisition, Validation, Investigation, Visualization, Methodology, Writing—original draft, Project administration, Writing—review and editing; Julia W Metzger, Data curation, Formal analysis, Funding acquisition, Validation, Investigation, Visualization, Methodology, Writing—original draft, Writing—review and editing; Talmo D Pereira, Data curation, Formal analysis, Validation, Investigation, Visualization, Methodology, Writing—review and editing; Thomas J Pisano, Data curation, Software, Formal analysis, Validation,

Visualization, Methodology, Writing—review and editing; Ben Deverett, Dariya E Bakshinskaya, Data curation, Software, Formal analysis, Visualization, Methodology, Writing—review and editing; Samuel S-H Wang, Formal analysis, Investigation, Writing—review and editing

## Author ORCIDs
Aleksandra Badura (iD) http://orcid.org/0000-0002-0119-5108
Jessica L Verpeut (iD) http://orcid.org/0000-0003-2941-7697
Julia W Metzger (iD) http://orcid.org/0000-0002-6380-8141
Talmo D Pereira (iD) http://orcid.org/0000-0001-9075-8365
Ben Deverett (iD) http://orcid.org/0000-0002-3119-7649
Samuel S-H Wang (iD) http://orcid.org/0000-0002-0490-9786

## Ethics

Animal experimentation: Experimental procedures were approved by the Princeton University Institutional Animal Care and Use Committee (protocol number: 1943-16) and performed in accordance with the animal welfare guidelines of the National Institutes of Health. All mice were housed in Optimice cages (Animal Care Systems, Centennial, CO) containing blended bedding (The Andersons, Maumee, OH), paper nesting strips, and one heat-dried virgin pulp cardboard hut (Shepherd Specialty Papers, Milford, NJ). PicoLab Rodent Diet food pellets (LabDiet, St. Louis, MO) and drinking water (or CNO water in the developmental groups) were provided ad libitum. Mice were relocated to clean cages with new component materials every two weeks. All mice were group-housed in reverse light cycle to promote maximal performance during behavioral testing. All surgery was performed under isoflurane anesthesia (5% for induction, 1-2% in oxygen; 1 L/min) and daily monitoring was employed to minimize suffering.

## Decision letter and Author response
Decision letter https://doi.org/10.7554/eLife.36401.036
Author response https://doi.org/10.7554/eLife.36401.037

## Additional files

### Supplementary files
• Transparent reporting form
DOI: https://doi.org/10.7554/eLife.36401.034

### Data availability

Code and data for the main figures are available via the GitHub repository https://github.com/wanglabprinceton/behavioral-development. The complete raw data for this study are available from the corresponding author upon request (including behavioral videos and serial two-photon tomographic brain images of each mouse).

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
