## [Decision Letter]

Thank you for submitting your article "Normal cognitive and social development require posterior cerebellar activity" for consideration by *eLife*. Your article has been reviewed by three peer reviewers, including Jennifer Raymond as the Reviewing Editor and Reviewer #3, and the evaluation has been overseen Richard Ivry as the Senior Editor. The following individuals involved in review of your submission have agreed to reveal their identity: Jörn Diedrichsen (Reviewer #2).

The reviewers have discussed the reviews with one another and the Reviewing Editor has drafted this decision to help you prepare a revised submission.

Summary:

This manuscript addressed an important topic in neuroscience, the role of the cerebellum in normal cognitive and social development. More specifically, this manuscript presents evidence that transient perturbation of cerebellar activity during postnatal development can have long-term effects on cognitive function. A chemogenetic approach was used to selectively perturb activity in four different cerebellar regions in juvenile mice, followed by behavioral testing in adult mice; and the results are compared with the effects of acute chemogenetic inactivation in adult mice during behavioral testing. This experimental study design is powerful. Although the cerebellum has been previously implicated in cognitive functions, and developmental perturbations have been used previously, the unique contribution of the present study design is the ability to compare transient inactivation of the cerebellum in juveniles versus adults. The results are likely to have significant scientific and translational impact. However, the presentation of the results could be significantly improved to make the manuscript clearer and stronger.

Essential revisions:

1) The core Results sections (Figure 2 and Figure 3) are complicated, and it is difficult to get a full overview of the results. Also, the authors should clarify which results are "punch line" results.

The PCA analysis and linear regression analysis only help here partially.

One possibility would be (a) z-standardize all behavioral scores of all groups and all task relative to the control group, with 0 meaning unimpaired (b) present a figure similar to the one in Figure 3E, but including both acute and developmental groups and the non-motor and motor test (locomotion and eye blink conditioning). This table will allow the reader to visually get a quick overview over the results – especially if you mark significant deficits.

In addition, each of the behavioral tasks is worthy of its own main figure or two (there is no limit on number of figures at *eLife*), with all of the relevant controls, most importantly, the direct comparison of adult vs juvenile inactivation of a given cerebellar region with appropriate statistical comparisons of the two (see point #2).

2) In its present form, the presentation of the results does not leverage the power of the experimental design. The main interesting contrast – the one between developmental and adult perturbation – is strangely under-emphasized. The results should be presented to facilitate and focus on the direct comparison of juvenile vs adult inactivation of the same cerebellar region on each of the behavioral tasks.

The direct developmental-acute contrast is reported only sometimes, and it is very difficult for the reader to get a full impression of similarities and differences. The reader should not have to go back and forth between main figures and supplementary to try to cobble together this key comparison. Moreover, after investing the effort to do so, it was in some cases still difficult to know what to conclude. For example, the controls' performance on open arm preference in Figure 2I va Figure 2—figure supplement 2C varies by almost twofold, which is similar to the effect sizes for cerebellar inactivation, which the authors highlight. This makes one wonder how to interpret the results. Does the difference between the two control groups arise from testing at different ages? If so, then this needs to be addressed in comparing the results of juvenile vs acute cerebellar inactivation.

Some analysis of how equivalent the juvenile and adult manipulations are in terms of their localization and amplitude is needed. Since the injections of virus were done at different ages, it would be helpful for the authors to provide a direct comparison of where the DREADDs are expressed after the juvenile vs adult injections, including an analysis of which cell types express the DREADs. This is especially important since the effect of adult acute inactivation of lobule VII (Figure 2—figure supplement 2B) looks very much like the effects of juvenile inactivation of Crus 1 (Figure 2F), and the histology shown in Figure 3D suggests that those two types of injections are, in aggregate, overlapping.

A full overview over the results (see comment #1) will help here – but as an additional analysis you could test how similar the full functional profiles (test-scores across all motor and non-motor tests) are between each developmental and control mouse (as compared to the within-group correlations). Either way, a clear discussion of the possible lasting cortical or cerebellar changes caused by a developmental perturbation is needed.

3) The principle component analysis and its conclusions are not very clear. The main problem with the current analysis is that it is performed for each task separately and does not include the motor measures and therefore fails to give a global overview of the data. Secondly, the author seems to rely on the hope that the principle component analysis will somehow find useful and semantically meaningful directions in the behavioral profile from the variation in the control mice. It is not clear that this hope is substantiated. If one of the PCs reflects motor behaviors only, should it not be impaired clearly by one of the lesions? The authors should consider removing this analysis in favor of a careful visual display of the global results, as suggested above. The PCA analysis can still be used for visualization as in the scatterplots in Figure 3B.

4) The negative weights in the linear regression analysis are puzzling. Do these mean that the experimental group performed better than the control? This depends on a number of things: Did you include the values for the control group into the regression? Did you include an intercept? Did you standardize scores before? Should the linear regression weights not show a very similar pattern from the significant group effects – it seems from Figure 1E that only crus I and the social chamber tasks are significant? There are currently simply not enough details given about the analysis to make sense of the results.

5) Throughout the text you seem to draw analogies between "homologous" regions in the mouse and human – based on lobular annotation. Looking at what we know about the lobular connectivity patterns in both species, the homologies (which are tricky to establish anyway) cannot necessarily be established on a lobules-by-lobules basis. Also, the connectivity pattern in humans does not really respect lobular boundaries if you inspect the Buckner et al., (2011) results carefully. Therefore, it is very likely that connectivity will have shifted substantially in evolution – and your tracing results (while showing specificity) do seem to confirm this. The current study provides absolutely valuable insights into the role of cerebellar circuits in development and adulthood – but mouse-human analogies should be discussed with more care.

6) The generalization from specific measurements on specific behavioral tasks to broad cognitive categories like novelty seeking and sociability is excessive. A subtle effect on one behavioral assay does not necessarily mean that the effect is truly the result of the label the investigators put on that assay e.g., a social deficit vs some other specific feature of the task. The results of the sociability test in Figure 2F suggest that the biggest effect is not a decrease in time with the other mouse, but rather an increase in time with the object, which one might equally be labelled a broadening of curiosity rather than a decrease in sociability. There are many additional places in the manuscript where the labeling of specific measurements on specific assays seemed too loose, and too contrived to make a case about autism (elevated plus is often described as measuring anxiety, but here we get a whole new set of cognitive labels from a single assay). Fine to do some speculation in the Discussion section, but more restraint and rigor in the interpretation would strengthen the manuscript.

[Editors' note: further revisions were requested prior to acceptance, as described below.]

Thank you for submitting your article "Normal cognitive and social development require posterior cerebellar activity" for consideration by *eLife*. Your article has been reviewed by three peer reviewers, including Jennifer L Raymond as the Reviewing Editor and Reviewer #3, and the evaluation has been overseen Richard Ivry, the Senior Editor. The following individuals involved in review of your submission have agreed to reveal their identity: Akira Sawa (Reviewer #1); Jörn Diedrichsen (Reviewer #2).

The reviewers have discussed the reviews with one another and the Reviewing Editor has drafted this decision to help you prepare a revised submission.

Summary:

This manuscript addressed an important topic in neuroscience, the role of the cerebellum in normal cognitive and social development. More specifically, this manuscript presents evidence that transient perturbation of cerebellar activity during postnatal development can have long-term effects on cognitive function. A chemogenetic approach was used to selectively perturb activity in four different cerebellar regions in juvenile mice, followed by behavioral testing in adult mice; and the results are compared with the effects of acute chemogenetic inactivation in adult mice during behavioral testing. This experimental study design is powerful. Although the cerebellum has been previously implicated in cognitive functions, and developmental perturbations have been used previously, the unique contribution of the present study design is the ability to compare transient inactivation of the cerebellum in juveniles versus adults.

In response to the previous review, the authors have significantly reorganized the presentation of the results in a way that increases clarity tremendously. The experiments are well controlled and rigorously analyzed. The systematic comparison of {different behavioral tasks and behavioral metrics} X {developmental time point for the perturbation} X {regional variation} provides a rich and valuable data set. The finding of different effects of developmental vs adult perturbations is novel and should be of broad interest. The reviewers have only two remaining concerns.

Essential revisions:

1) In response to the reviewers' previous query about negative weights in the linear regression, the authors now use absolute value. This is a step in the wrong direction, as it obscures whether more lesion in the structure improved or impaired function. A better approach may be to show the signed value, but flip the sign, so that positive values consistently (across behavioral measures) indicate impairments and negative values improvement in function (or the other way around). Again, removing the sign and therefore this crucial information did not address the reviewers' concern.

Related to this, the questions about inclusion of control group and intercept were not answered clearly. One might guess that the control group was not included, but all 4 injection groups, separately for the adult and juvenile cohorts? Please clarify.

---

## [Author Response]

We thank the reviewers for their careful attention to our manuscript. Presenting a large and multidimensional body of experimental work is a challenge. The reviewers’ suggested revisions are a substantial improvement.

We have edited the manuscript and reorganized the figures to address the points raised by the reviewers. We now present acute and developmental cerebellar perturbations side by side. We also discuss all differences in the behavioral tests instead of highlighting just a few. Finally, we include a table where we summarize the results

Our point-by-point response to comments is given below.

Essential revisions:1) The core Results sections (Figure 2 and Figure 3) are complicated, and it is difficult to get a full overview of the results. Also, the authors should clarify which results are "punch line" results.The PCA analysis and linear regression analysis only help here partially.One possibility would be (a) z-standardize all behavioral scores of all groups and all task relative to the control group, with 0 meaning unimpaired (b) present a figure similar to the one in Figure 3E, but including both acute and developmental groups and the non-motor and motor test (locomotion and eye blink conditioning). This table will allow the reader to visually get a quick overview over the results – especially if you mark significant deficits.

We now summarize principal results in units of effect size in Table 3.

In addition, each of the behavioral tasks is worthy of its own main figure or two (there is no limit on number of figures at eLife), with all of the relevant controls, most importantly, the direct comparison of adult vs juvenile inactivation of a given cerebellar region with appropriate statistical comparisons of the two (see point #2).

We have now rearranged the figures in this manner.

2) In its present form, the presentation of the results does not leverage the power of the experimental design. The main interesting contrast – the one between developmental and adult perturbation – is strangely under-emphasized. The results should be presented to facilitate and focus on the direct comparison of juvenile vs adult inactivation of the same cerebellar region on each of the behavioral tasks.The direct developmental-acute contrast is reported only sometimes, and it is very difficult for the reader to get a full impression of similarities and differences. The reader should not have to go back and forth between main figures and supplementary to try to cobble together this key comparison. Moreover, after investing the effort to do so, it was in some cases still difficult to know what to conclude. For example, the controls' performance on open arm preference in Figure 2I va Figure 2—figure supplement 2C varies by almost twofold, which is similar to the effect sizes for cerebellar inactivation, which the authors highlight. This makes one wonder how to interpret the results. Does the difference between the two control groups arise from testing at different ages? If so, then this needs to be addressed in comparing the results of juvenile vs acute cerebellar inactivation.

We now place adult/developmental data together in the same figure, and directly next to one another as a matched pair as appropriate.

Some analysis of how equivalent the juvenile and adult manipulations are in terms of their localization and amplitude is needed. Since the injections of virus were done at different ages, it would be helpful for the authors to provide a direct comparison of where the DREADDs are expressed after the juvenile vs adult injections, including an analysis of which cell types express the DREADs. This is especially important since the effect of adult acute inactivation of lobule VII (Figure 2—figure supplement 2B) looks very much like the effects of juvenile inactivation of Crus 1 (Figure 2F), and the histology shown in Figure 3D suggests that those two types of injections are, in aggregate, overlapping.

In all cases, expression was confined to molecular layer interneurons (See New Figure 1D and Video 1). We have added a mention of this to the abstract.

We fully agree that it is necessary to address the fact that injection regions overlap. This was the rationale for the linear model. We now mention the linear model in the Abstract and explain it at greater length in the Results section.

A full overview over the results (see comment #1) will help here – but as an additional analysis you could test how similar the full functional profiles (test-scores across all motor and non-motor tests) are between each developmental and control mouse (as compared to the within-group correlations). Either way, a clear discussion of the possible lasting cortical or cerebellar changes caused by a developmental perturbation is needed.

In response to this comment, we attempted to visualize the mouse-to-mouse variability using multidimensional scaling. However, the results did not grab the eye. One likely reason is the variability of DREADD expression.

In our view, this point is made most powerfully by the linear model. We have expanded that figure to include example scatterplots of the relationships that go into the linear model. These capture both, the variability and the nature of the functional contribution of each lobule.

3) The principle component analysis and its conclusions are not very clear. The main problem with the current analysis is that it is performed for each task separately and does not include the motor measures and therefore fails to give a global overview of the data. Secondly, the author seems to rely on the hope that the principle component analysis will somehow find useful and semantically meaningful directions in the behavioral profile from the variation in the control mice. It is not clear that this hope is substantiated. If one of the PCs reflects motor behaviors only, should it not be impaired clearly by one of the lesions? The authors should consider removing this analysis in favor of a careful visual display of the global results, as suggested above. The PCA analysis can still be used for visualization as in the scatterplots in Figure 3B.

For simpler presentation, we now use within-group variance of single metrics. We have changed the figures to focus on this rather than PCA.

It is true that none of our lesions impaired PC1, which captures the largest share of behavioral variation in control groups. This is in fact the notable observation: PC1 does not capture the variability within a treatment group. In our view this is a rigorous way to test the idea that perturbations do something unusual to mice.

4) The negative weights in the linear regression analysis are puzzling. Do these mean that the experimental group performed better than the control? This depends on a number of things: Did you include the values for the control group into the regression? Did you include an intercept? Did you standardize scores before? Should the linear regression weights not show a very similar pattern from the significant group effects – it seems from Figure 1E that only crus I and the social chamber tasks are significant? There are currently simply not enough details given about the analysis to make sense of the results.Negative coefficients mean that more expression is associated with a lower score on that metric. If positive values indicate more function, a negative coefficient means there is impairment. To make things clearer, we now display absolute value. The direction of change is shown elsewhere.5) Throughout the text you seem to draw analogies between "homologous" regions in the mouse and human – based on lobular annotation. Looking at what we know about the lobular connectivity patterns in both species, the homologies (which are tricky to establish anyway) cannot necessarily be established on a lobules-by-lobules basis. Also, the connectivity pattern in humans does not really respect lobular boundaries if you inspect the Buckner et al., (2011) results carefully. Therefore, it is very likely that connectivity will have shifted substantially in evolution – and your tracing results (while showing specificity) do seem to confirm this. The current study provides absolutely valuable insights into the role of cerebellar circuits in development and adulthood – but mouse-human analogies should be discussed with more care.

In the Introduction we now emphasize that the anteroposterior lobular boundaries are useful as anatomical landmarks, but not necessarily functional boundaries. We now clarify that microzones are the established organizing principle of cerebellar cortex.

6) The generalization from specific measurements on specific behavioral tasks to broad cognitive categories like novelty seeking and sociability is excessive. A subtle effect on one behavioral assay does not necessarily mean that the effect is truly the result of the label the investigators put on that assay e.g., a social deficit vs some other specific feature of the task. The results of the sociability test in Figure 2F suggest that the biggest effect is not a decrease in time with the other mouse, but rather an increase in time with the object, which one might equally be labelled a broadening of curiosity rather than a decrease in sociability. There are many additional places in the manuscript where the labeling of specific measurements on specific assays seemed too loose, and too contrived to make a case about autism (elevated plus is often described as measuring anxiety, but here we get a whole new set of cognitive labels from a single assay). Fine to do some speculation in the Discussion section, but more restraint and rigor in the interpretation would strengthen the manuscript.

We now specify parameter name in the social assay to clarify what measured quantity actually changed. We save our most speculative thoughts for the Discussion section.

[Editors' note: further revisions were requested prior to acceptance, as described below.]Essential revisions:1) In response to the reviewers' previous query about negative weights in the linear regression, the authors now use absolute value. This is a step in the wrong direction, as it obscures whether more lesion in the structure improved or impaired function. A better approach may be to show the signed value, but flip the sign, so that positive values consistently (across behavioral measures) indicate impairments and negative values improvement in function (or the other way around). Again, removing the sign and therefore this crucial information did not address the reviewers' concern.

We have restored the signed information, and redefined the sign of each behavioral measure so that lost function is indicated as a positive value. We now define the sign conventions for impairments to Table 2. This information is reflected in Table 3 (and its supplement) and Figure 10.

Related to this, the questions about inclusion of control group and intercept were not answered clearly. One might guess that the control group was not included, but all 4 injection groups, separately for the adult and juvenile cohorts? Please clarify.

The linear regression model that we are presenting correlates the phenotypic scores with the amount of expression. Given that the control groups have no expression only experimental (injected) groups were used. Thus we regressed the behavioral scores of 4 injection groups, separately for the adult and juvenile cohorts, onto their respective expression profiles. An intercept was included in the fit. The fraction of voxels that had expression can be seen for each mouse in Figure 10—figure supplement 1. We have clarified that in the Results section and in the Materials and methods section.

We have restored the signs of the regression weights as requested. In addition, we colored Table 3 (and its supplement) and Figure 10 so that red indicates impairment in function and blue an improvement.